# Fingertip viscoelasticity enables human tactile neurons to encode loading history alongside current force

Hannes P Saal[1]*, Ingvars Birznieks[2,3], Roland S Johansson[4]

[1]Active Touch Laboratory, School of Psychology, University of Sheffield, Sheffield, United Kingdom; [2]School of Biomedical Sciences, UNSW Sydney, Sydney, Australia; [3]Neuroscience Research Australia, Sydney, Australia; [4]Physiology Section, Department of Integrative and Medical Biology, Umeå University, Umeå, Sweden

### eLife Assessment

The **fundamental** findings reported here provide insight into how the viscoelasticity of the fingertip skin influences the activity of mechanoreceptive afferents and thus the neural coding of force in humans. The basic principle studied was whether and to what extent the previous applied force directions impact the firing of FA-1, SA-1 and SA-2 neurons during the current applied force directions. The data and analyses are **compelling** and will be helpful for modeling the neural representations of force in the context of object grasping and manipulation.

**\*For correspondence:**
h.saal@sheffield.ac.uk

**Competing interest:** The authors declare that no competing interests exist.

**Abstract** Human skin and its underlying tissues constitute a viscoelastic medium, implying that any deformation depends not only on the currently applied force, but also on the recent loading history. The extent to which this physical memory influences the signaling of first-order tactile neurons during natural hand use is not well understood. Here, we examined the effect of past loading on the responses of fast-adapting (FA-1) and slowly-adapting (SA-1 and SA-2) first-order tactile neurons innervating the human fingertip to loadings applied in different directions representative of object manipulation tasks. We found that variation in the preceding loading affected neurons' overall signaling of force direction. Some neurons kept signaling the current direction, while others signaled both the current and preceding direction, or even primarily the preceding direction. In addition, ongoing impulse activity in SA-2 neurons between loadings signaled information related to the fingertip's viscoelastic deformation state. We conclude that tactile neurons at the population level signal continuous information about the fingertip's viscoelastic deformation state, which is shaped by both its recent history and current loading. Such information might be sufficient for the brain to correctly interpret current force loading and help in computing accurate motor commands for interactions with objects in manipulation and haptic tasks.

## Introduction

To enable successful skilled object manipulation and haptic object exploration, the brain must have access to information related to the forces acting on the fingertips (*Johansson and Westling, 1984*; *Westling and Johansson, 1984*; *Robles-De-La-Torre and Hayward, 2001*). Experimental evidence indicates that populations of first-order tactile neurons with end-organs distributed throughout the fingertip skin provide sensory information about the distribution, magnitude, and direction of such fingertip forces (see *Johansson, 2008*; *Johansson and Flanagan, 2009*, for reviews). However, these neurons do not signal contact forces per se, but local tissue deformations at the site of their receptor

organs. This means that the relationship between fingertip forces and neuronal signaling can be very complex, since the deformation patterns resulting from a force applied to the fingertip depend on its geometry and its non-linear, viscoelastic, and anisotropic material properties (*Serina et al., 1997*; *Pawluk and Howe, 1999*; *Nakazawa et al., 2000*; *Jindrich et al., 2003*; *Pataky et al., 2005*; *Wang and Hayward, 2007*). Concerning the viscoelastic properties in particular, the recent loading history of the fingertip might influence a neuron's response to a given contact force. That is, since tissue viscoelasticity causes deformation changes to lag force changes, residual deformations from previous loadings will affect how the fingertip reacts mechanically under a given loading, and thus affect the deformation changes to which neurons' receptor organs are exposed.

Despite indications in early animal studies that stimulation history via tissue viscoelasticity may indeed affect the responsiveness of first-order tactile neurons (*Lindblom, 1965*; *Werner and Mount-castle, 1965*; *Beitel et al., 1977*; *Pubols, 1982*), the issue has subsequently received little attention in studies of tactile mechanisms. Because the viscoelasticity of human fingertips exhibits time constants of up to several seconds (*D'Angelo et al., 2016*; *Kumar et al., 2015*; *Pawluk and Howe, 1999*; *Wu et al., 2003*), we hypothesized that loading history would interfere with the signaling of first-order tactile neurons to rapidly fluctuating fingertip forces on a similar time scale to those experienced naturally (*Kunesch et al., 1989*; *Callier et al., 2015*; *Morley et al., 1983*).

Here, we tested the effect of loading history on neural information transmission in human first-order tactile neurons about the direction of fingertip forces during repetitive loadings that mimic those occurring during natural object manipulation. We examined information conveyed in the three types of neurons: fast-adapting type I (FA-1), slowly adapting type I (SA-1), and slowly adapting type II (SA-2; *Birznieks et al., 2001*; *Johansson and Birznieks, 2004*; *Birznieks et al., 2009*; *Saal et al., 2009*). These neuron types most likely supply Meissner corpuscles, Merkel cell neurite complexes, and Ruffini-like end-organs, respectively. For neurons of all three types, we show that variations in loading history not only affect fingertip deformation but also reduce information about the force direction in the prevailing loading. Although most neurons of each type continued to preferentially signal information about the current force direction, a minority signaled more information about the preceding force direction than the current one. For those SA-2 neurons that exhibit ongoing activity

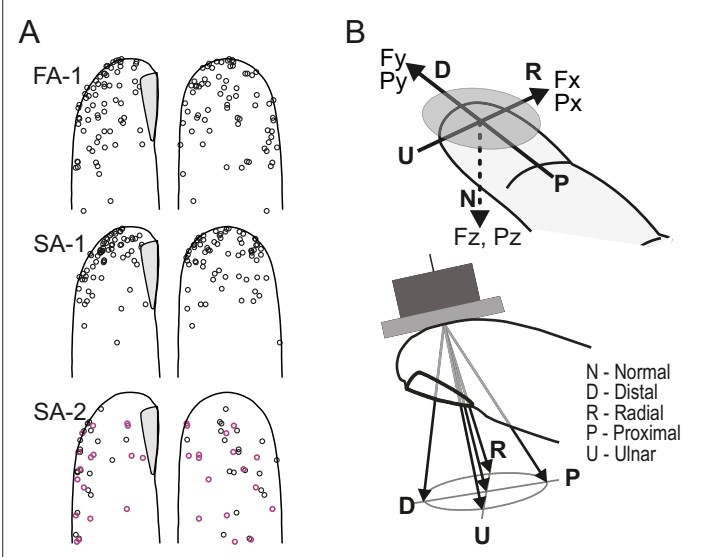

**Figure 1.** Experimental setup. (**A**) Receptive field center locations shown on a standardized fingertip for all first-order tactile neurons included in the study, categorized by neuron type. Purple symbols denote spontaneously active SA-2 neurons exhibiting ongoing activity without external stimulation. (**B**) The flat stimulus surface was centered at the standard site of stimulation and oriented such that its tangential plane was parallel to the flat portion of skin on the fingertip. The stimulus maintained contact with the skin at a force of 0.2 N in intertrial periods. Force stimuli were superimposed on this background contact force and were delivered in the normal direction (N) and at an angle of 20 degrees to the normal with tangential components in the distal (D), radial (R), proximal (P), and ulnar (U) directions, as indicated by the five arrows in the lower panel.

without external stimulation (*Birznieks et al., 2009*; *Johansson, 1978*; *Knibestöl, 1975*; *Chambers et al., 1972*), we found that they could signal information about the viscoelastic state of the fingertip even when unloaded.

## Results

We recorded action potentials in the median nerve of individual low-threshold A-beta myelinated first-order human tactile neurons innervating the glabrous skin of the fingertip (*Vallbo and Hagbarth, 1968*). Sixty of the neurons were fast adapting type I (FA-1), 73 were slowly adapting type I (SA-1), and 41 were slowly adapting type II (SA-2) neurons (*Vallbo and Johansson, 1984*). The receptive fields of the neurons within each class were widely distributed over the glabrous skin of the distal phalanx (*Figure 1A*). The fourth type of tactile neurons in the human glabrous skin, fast-adapting type II neurons (FA-2) supplied by Pacinian corpuscles, were not considered because our stimuli did not contain mechanical events with frequency components high enough to reliably excite them. Fingertip forces were applied in five different directions with a flat surface (referred to as the contactor), which was always oriented parallel to the skin surface at the primary site of object contact in dexterous tasks, that is in the middle of the flat portion of the volar surface of the fingertip. Because forces were applied to a standardized site, the neurons could vary widely in their responsiveness depending on where in the mechanically complex fingertip their transduction sites were located. Forces were applied normal (N) to the skin and at 20 degrees to the normal direction in the radial (R), distal (D), ulnar (U), or proximal (P) direction, respectively (*Figure 1B*). All force stimuli consisted of a force protraction phase (125ms), a plateau phase at 4 N normal force (250ms), and a force retraction phase (125ms); interstimulus periods were 250ms.

Each neuron was subject to two different sequences of force stimuli, first the 'regular sequence' and then the 'irregular sequence'. As stimuli were force controlled, the contactor's movement and position producing identical reactive force may differ depending on stimulation history, due to the viscoelastic

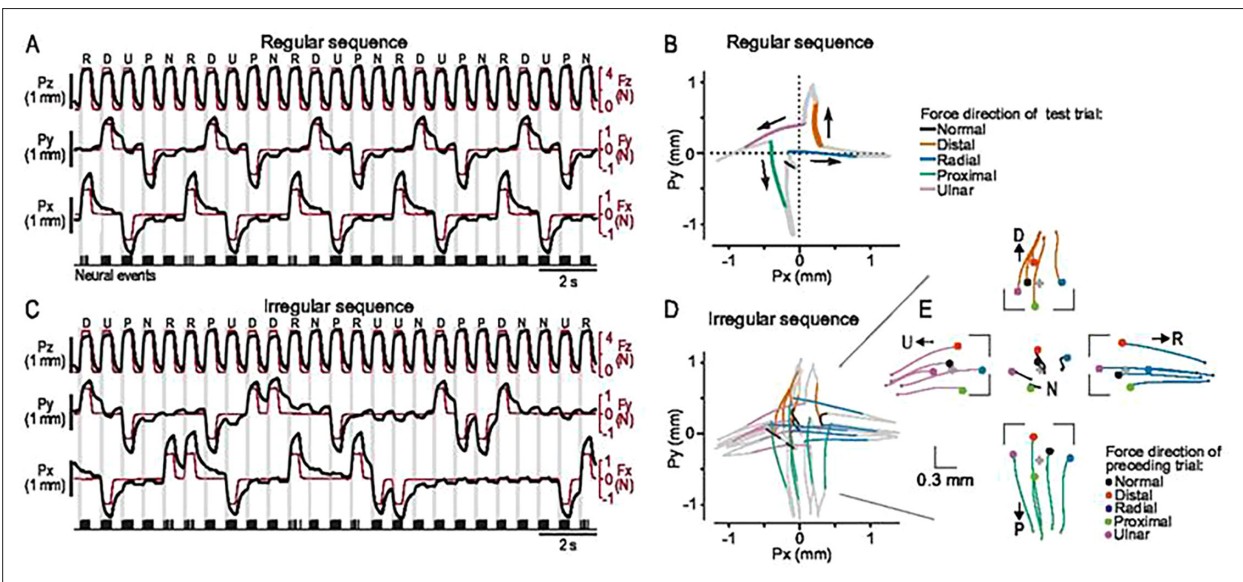

**Figure 2.** Stimulation sequence exposes fingertip viscoelasticity. (**A**) Trial order for the entire regular sequence, which repeats fingertip loadings in five different force directions in a fixed order, implying that loadings in each direction received the same stimulation history. Force (red lines) and contactor position (black lines) are shown along the normal (z), distal/proximal (y), and ulnar-radial (x) axes, while recording action potentials from a single exemplary SA-1 neuron (bottom trace). (**B**) Average contactor position in the tangential plane for all trials in the regular sequence across all recorded neurons (and thus fingertips). The colored segments of the curves indicate the protraction phase for each of the five force directions, while other phases of the fingertip loading (plateau, retraction) and the interstimulus period are shown in grey. Dashed lines show the directions in which the tangential force components were applied. (**C**) Trial order for the entire irregular sequence, where force directions are varied such that trials in each stimulation direction were preceded by loading in each of the five directions once. Same neuron (and fingertip) as in A. (**D**) Average contactor position in the tangential plane for the irregular sequence. Same format as in B. (**E**) Contactor position in the tangential plane at the start of (filled circles) and during the force protraction phase (colored lines) per force direction, referenced to the fingertip position at rest (gray marker). Same trials as in D, but different force directions are shown in separate panels for better visibility.

properties of the fingertip tissue. In the regular sequence, the five force directions were repeatedly presented in a fixed relative order (R, D, U, P, N), such that each loading in each direction received the same immediate stimulation history (*Figure 2A*). Differences in the path of the contactor between different force directions were clearly visible in the tangential plane, where the changes in the force stimulation took place between trials (*Figure 2B*). In the regular sequence, the contactor path was practically identical across the five test trials in each direction (*Figure 2B*), indicating that the fingertip deformed similarly during the repetitions. However, the contactor path deviated from the direction of the force due to anisotropic mechanical properties of the fingertip, and, importantly, it differed between force protraction and retraction due to the viscoelasticity of the fingertip (*Figure 2B*). Likewise, the viscoelastic properties were reflected as a pronounced hysteresis between the force and the contactor position, creep during the force plateaus, and creep recovery during the interstimulus periods (*Figure 2A*).

In the irregular sequence, also including five trials in each force direction, the stimulation history for trials in each direction varied systematically, such that trials in each stimulation direction were preceded by loading in each of the five directions once (*Figure 2C*). As in the regular sequence, the contactor moved broadly in the direction of the force regardless of the previous loading direction, but its precise path differed markedly depending on the previous loading direction (*Figure 2D*). This variability was most evident during the protraction phase (*Figure 2E*) but could be discerned throughout the trial (*Figure 2D*).

Hence, deduced from the contactor's behavior, these observations indicated that variations in the immediate loading history caused a greater intertrial variation in the fingertip deformation than an unchanging loading history. This is illustrated in *Figure 3A*, which shows the contactor path for both regular and irregular sequences from a single participant for five test trials in the distal direction as well as the corresponding preceding trials.

## Effects of loading history on neural responses

We asked whether the greater intertrial variability in fingertip deformation during the irregular stimulation sequence would be reflected in the responses of tactile neurons. Indeed, we observed greater intertrial variability in the firing rate profiles of the neurons in the irregular compared to the regular sequence during test trials (see examples in *Figure 3B–D*). This increased variability was most evident during the force protraction phase, where most neurons exhibited the most intense responses.

Increased variability was also observed in instances where the dynamic response to force stimulation involved a decrease in the firing rate (lower panels of *Figure 3D*). This phenomenon was observed in SA-2 neurons that maintained an ongoing discharge during intertrial periods (*Figure 1A*). In these cases, the response to a force stimulus constituted a modulation of the firing rate around the background discharge, signifying that a force stimulus could either decrease or increase the firing rate depending on the prevailing stimulus direction. For SA-1 and SA-2 neurons, which typically generated nerve impulses also during the force plateau, and for FA-1 neurons, which often responded during force retraction, we noted a tendency for greater variability in the intensity of these responses as well. Finally, the variation in previous loading direction during the irregular sequence could also modify impulse activity in SA-2 neurons generated during interstimulus periods, the implications of which will be further addressed below.

We quantified the effect of the stimulation history on a neuron's response by first calculating for each stimulation sequence the time-varying standard deviation of the instantaneous firing rate during the test trials in each of the five force directions. Averaged across neurons and loading directions, firing rate variability during the force protraction phase was just over twofold higher for FA-1s in the irregular sequence compared to the regular sequence, almost twofold higher for SA-1 and about 70% higher for SA-2 neurons (*Figure 4A*). In addition to the protraction phase, variability was elevated in the irregular sequence for both SA types during the plateau phase and for FA-1 neurons during the retraction phase (all $p_{corrected} <= 0.001$, paired Wilcoxon signed rank tests). FA-1 neurons did not respond during the plateau phase (see *Figure 4B*), while both SA types responded only weakly during the retraction phase with no apparent difference between the sequences ($p >= 0.3$). Thus, in all phases where neurons responded reliably, their response variability in test trials increased when the force direction in the preceding trial varied. Notably, neurons' overall firing rates did not differ between the regular and irregular sequence when averaged over test trials in all force directions (*Figure 4B*) other

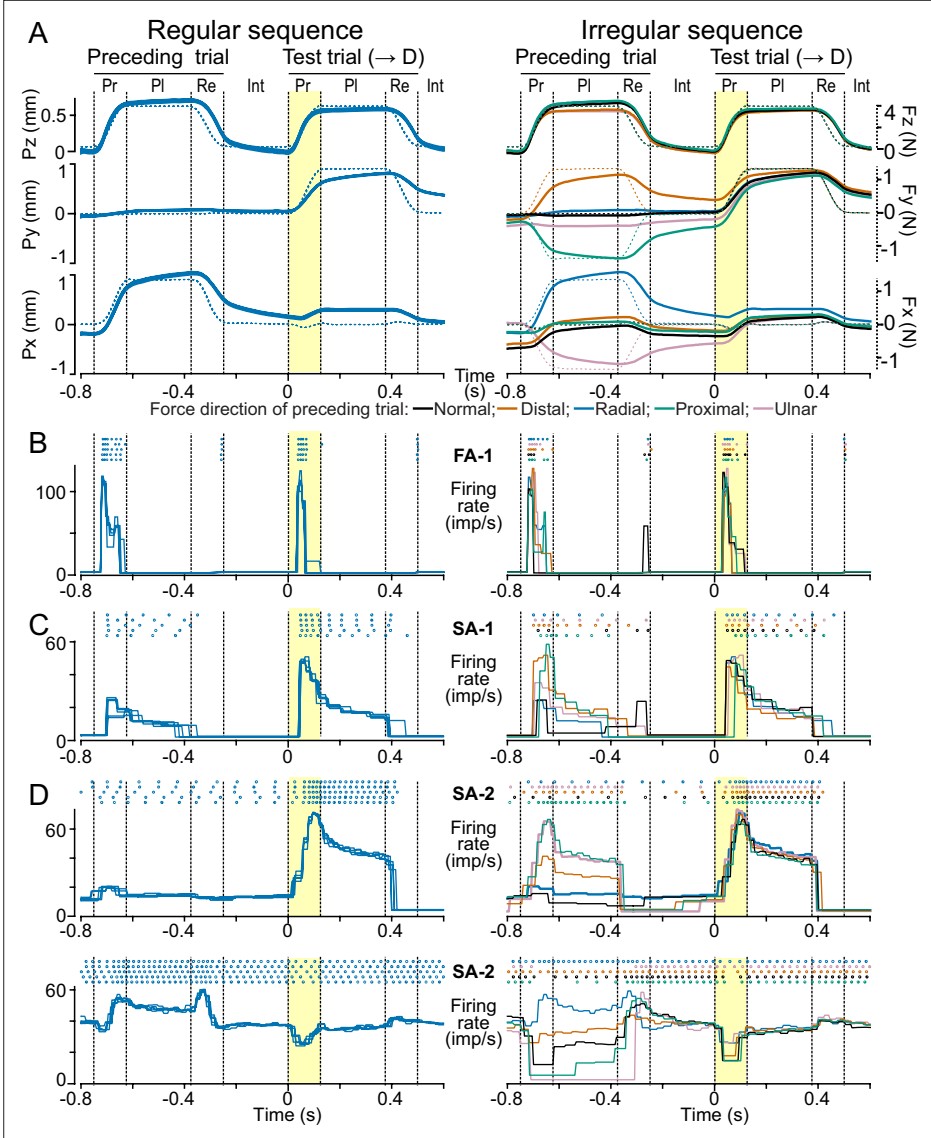

**Figure 3.** Influence of preceding loading direction on fingertip deformation and neural responses. (**A**) Contactor position along the x, y, and z axes (see *Figure 1B*) as a function of time superimposed for all five trials with loading in the distal direction ('test trial') as well as the respective previous loading ('preceding trial') in the regular (left column) and the irregular (right column) sequence. Vertical dashed lines mark transitions between loading phases (Pr: protraction, Pl: plateau, Re: retraction phase, Int: interstimulus period). The yellow shaded area indicates the protraction phase of the test trials. Each trace is colored according to the force direction of the previous loading. Data were recorded from a neuron whose response is shown in B. (**B**) Dots (top) represent action potentials recorded from an FA-1 neuron for each trial, whose contactor movements are shown in A. The superimposed traces below represent the corresponding firing rate profiles, defined as the reciprocal of the interval between subsequent action potentials. Color coding as in A. (**C, D**) Exemplary responses of one SA-1 (**C**) and two SA-2 neurons (**D**) to force loadings corresponding to those in A. All neurons show higher variability in their firing rate profiles during test trials in the irregular compared to the regular sequence.

than for SA-1s during the protraction phase ($p_{corrected}$ = 0.002), where the difference in firing rate was less than 1 imp/s. This suggested that the greater variability was linked to the stimulation history and not a change in the neurons' overall responsiveness.

We also analyzed the effect of stimulation sequence on the fingertip deformation, again by using the contactor behavior as a proxy. For each neuron examined and stimulation sequence, we calculated the time-varying standard deviation in the tangential plane of the contactor position and of its

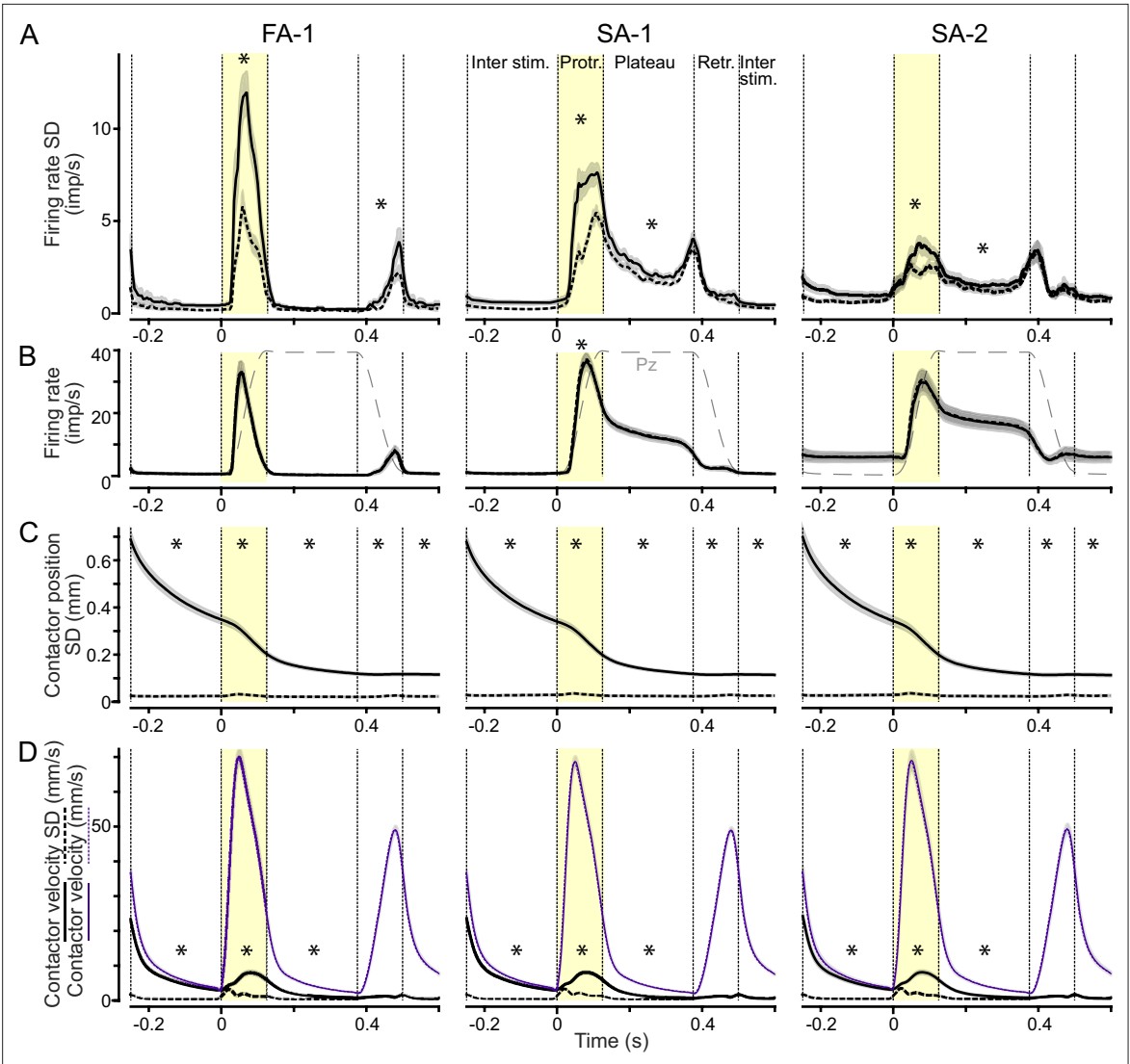

**Figure 4.** Increased variability in neural responses and fingertip deformations in the irregular sequence. (**A**) Standard deviation of neuronal instantaneous firing rates as a function of time during test trials in each loading direction in the regular (dashed lines) and irregular (solid lines) sequences. Data averaged across neurons of each type and all loading directions. Shaded areas indicate SEM. Vertical dashed lines mark transitions between loading phases as in **Figure 3A** and stars indicate phases where there was a significant difference between regular and irregular sequences at $p < 0.05$. (**B**) Average instantaneous firing rates for the regular and irregular sequence as a function of time. Dashed black lines indicate the force profile of the fingertip loading. Note that the average firing rates are almost identical in the regular and irregular sequence. (**C**) Standard deviation of tangential (2D) contactor position for the regular and irregular sequences. (**D**) Standard deviation of contactor velocities (black lines) and average contactor velocity (purple lines) for the regular and irregular sequences.

velocity across the five test stimuli in each of the five force directions. As results were similar across the different test directions, we then averaged these data to arrive at a single variability measure per neuron. Importantly, the variability in contactor position was markedly higher in the irregular than in the regular sequence (**Figure 4C**, solid vs. dashed lines). The variability in the irregular sequence decreased over time and did so especially rapidly during the force protraction phase. Averaged across all neurons, at the beginning of the force protraction, the variability in contactor position was about 20 times greater in the irregular than in the regular sequence. Variability was 7 times greater during the plateau phase and still about 4 times so at the end of the retraction phase. During all phases, including the interstimulus period before and after the test trial, the variability in the irregular sequence was significantly higher than in the regular sequence ($p_{corrected} < 0.001$, paired Wilcoxon signed rank tests). The effect of stimulation sequence present even at the end of the force retraction phase indicates that

the viscoelastic memory trace of the previous loading direction lingered to some extent even beyond the test trials. Notably, the time-varying variation in contactor position was virtually identical for data pertaining to each of the three neuron types (*Figure 4C*), demonstrating that the differences in neural response behavior could not be explained by variability in the skin responses across the different experimental runs.

Regarding the effect of viscoelasticity variability on contactor velocity, we found that the variability in contactor velocity was significantly greater in the irregular than in the regular sequence during the initial interstimulus period, the force protraction phase, and the plateau phase (*Figure 4D*, black curves; $p_{corrected} < 0.001$), but not during the retraction phase or the subsequent interstimulus period ($p > 0.4$). However, even during the irregular sequence, the variability in contactor speed appeared rather modest compared to the absolute contactor speed (*Figure 4D*, cf. black and gray curves), which likely primarily drove the dynamic neural response during the force protraction for all classes (*Figure 4B*).

Taken together, the variable loading history in the irregular sequence affected the neurons' firing rates, and most so during their dynamic responses elicited by the force protraction. Further, the effects on the neurons' responses reasonably matched the influence of the variation in previous force direction on the state of the fingertip deformation and its change during the test trials, which was also most pronounced during the force protraction phase.

## Predicting neural responses from contactor movements

The similarity in the history-dependent variation in neural firing and fingertip deformation at a given force stimulus suggests that neuronal firing is determined by how the fingertip deforms rather than the applied force itself. However, this similarity does not clarify the relationship between fingertip deformation dynamics and neural signaling. To investigate further, we fit cross-validated multiple linear regression models to evaluate how well distinct aspects of contactor movement could predict the time-varying firing rates of individual neurons during the protraction phases of the irregular sequence. The models used predictors based on (1) the three-dimensional position of the contactor, (2) its three-dimensional velocity, (3) a combination of position and velocity signals, and, finally, (4) position and

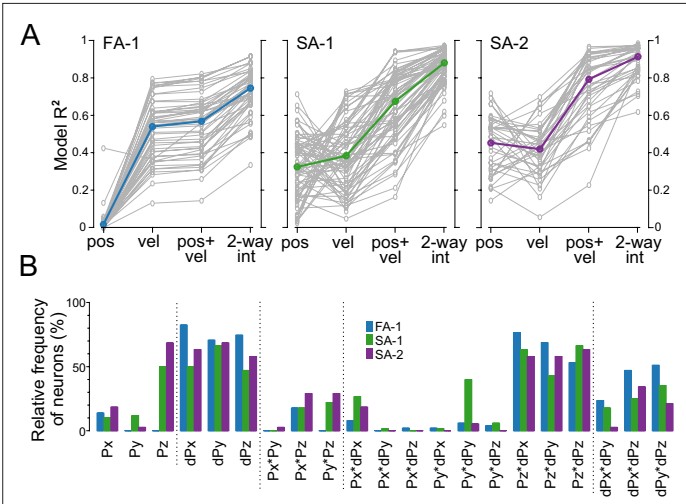

**Figure 5.** Neurons' sensitivity to different aspects of contactor movement. (**A**) Variance explained ($R^2$) when predicting time-varying firing rates using four regression models: (1) 3D contactor positions ('pos'), (2) 3D velocities ('vel'), (3) a combination of positions and velocities ('pos +vel'), and (4) a model incorporating all two-way interactions ('2-way int') for FA-1, SA-1, and SA-2 neurons. The interaction model consistently outperformed the others across all neuron types. (**B**) Relative frequency of neurons, categorized by type, for which a given predictor ranked among the six most important in the two-way interaction model. For a detailed breakdown of predictor importance by rank, see *Figure 5—figure supplement 1*.

The online version of this article includes the following figure supplement(s) for figure 5:

**Figure supplement 1.** Importance of individual predictors in the two-way interaction model.

velocity signals along with all possible two-way interactions between them, capturing potentially complex relationships between fingertip deformations and neural signaling.

Comparing the variance explained ($R^2$) by each regression model for each neuron type revealed clear differences between the models (*Figure 5A*). A two-way mixed design ANOVA, with regression model as within-group effects and neuron type as a between-group effect, revealed a main effect of model on variance explained ($F(3, 462) = 815.5$, $p < 0.001$, $\eta_p^2 = 0.84$). Model prediction accuracy overall increased with the number of predictors, with the two-way interaction model outperforming all others ($p < 0.001$ for all comparisons, Tukey's HSD). Additionally, a significant main effect of neuron type ($F(2, 154) = 29.8$, $p < 0.001$, $\eta_p^2 = 0.28$) and a significant interaction between regression model and neuron type were observed ($F(6, 462) = 50.8$, $p < 0.001$, $\eta_p^2 = 0.40$).

For neuron type, model predictions were most accurate for SA-2 neurons, followed by SA-1 neurons, with FA-1 neurons showing the lowest accuracy ($p < 0.003$ for all comparisons, Tukey's HSD). The interaction between model and neuron type revealed distinct patterns. For SA-1 and SA-2 neurons, position-only and velocity-only models had similar prediction accuracy ($p \geq 0.996$, Tukey's HSD) with no significant differences between these neuron types ($p \geq 0.552$, Tukey's HSD). FA-1 neurons performed poorly with the position-only model but showed higher accuracy with the velocity-only model ($p < 0.001$, Tukey's HSD) and better than SA-1 neurons ($p = 0.006$, Tukey's HSD). Models combining position and velocity predictors (without interactions) surpassed both position-only and velocity-only models for SA-1 and SA-2 neurons ($p < 0.001$, Tukey's HSD). Overall, the differences between neuron types broadly match their tuning to static and dynamic stimulus properties.

The two-way interaction model, accounting for most variance in neural responses, produced mean $R^2$ values of 0.75 for FA-1, 0.88 for SA-1, and 0.91 for SA-2 neurons (*Figure 5A*). To evaluate the contribution of the different predictors, we ranked them using the permutation feature importance method, focusing on the six most important ones. Regression analyses using only these variables explained almost all of the variance explained by the full model, with a median $R^2$ reduction of just 0.055 across all neurons. Across all neuron types, at least half included all three velocity components (dPx, dPy, dPz) among the top six, with FA-1 neurons showing the highest prevalence (*Figure 5B*). Interactions between normal position (Pz) and each velocity component were also frequently observed, while interactions involving tangential position and velocity components were less common. Interactions among velocity components were relatively well represented, followed by interactions limited to position components. Position signals were generally less represented, except for normal position (Pz) in slowly adapting neurons, where it appeared in 50% of SA-1 and 68% of SA-2 neurons. Despite these broad trends, important predictors varied widely across ranks even within a given neuron class (see *Figure 5—figure supplement 1*), and even the most frequent variables appeared in only a subset of cases, suggesting broad variability in sensitivity across neurons.

## Information transmission about past and present loading

To assess how the increased firing rate variability observed in the irregular sequence as well as the variation in sensitivity to different aspects of contactor movement affected neural information transmission, we calculated a lower bound on the mutual information transmitted about both the current and the preceding force direction based on the neural spiking responses of individual neurons (see Materials and methods). We focused on the protraction phase only, during which firing rates and their variability were highest.

Averaged across all neurons of each type, information about the current force direction tended to accumulate throughout the force protraction phase, as shown previously for FA-1 and SA-1 neurons (*Saal et al., 2009*). However, for all types, the rate of information increase was considerably lower and cumulative information tended to plateau at a much lower value in the irregular than in the regular sequence (compare black dashed with orange lines in *Figure 6A*). At the end of the protraction phase, both FA-1 and SA-1 neurons signaled on average only 50% of the information in the irregular compared to the regular sequence ($p < 0.001$ for both types, paired Wilcoxon signed rank tests). For the SA-2 neurons, the corresponding information transfer was reduced by only 20% ($p = 0.02$). However, these neurons signaled far less information about force direction to begin with. As a result, average information transmission in the three classes of neurons ended up comparable in the presence of viscoelastic effects.

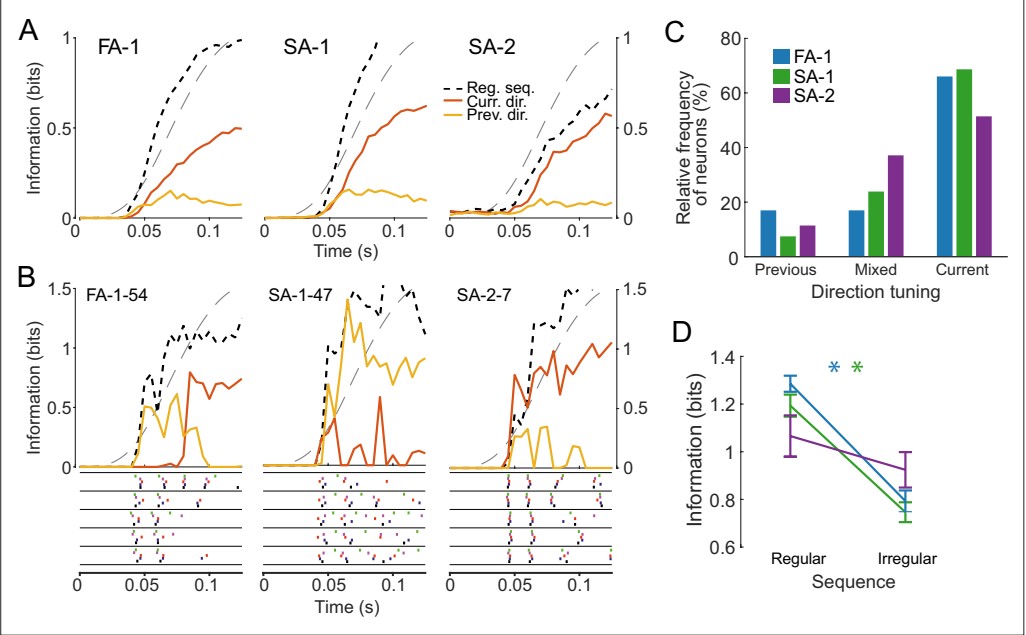

**Figure 6.** Information about current and previous force direction during the protraction phase. (**A**) Average mutual information about force direction for FA-1 (left), SA-1 (middle), and SA-2 (right) neurons as a function of time during the protraction phase in the irregular sequence. Information is shown for the current trial (solid orange line) or the preceding trial (solid yellow line) and is compared to the regular sequence (dotted black line). Gray dashed lines denote the stimulus force profile. (**B**) Mutual information curves (top) and spike trains (bottom) for three individual neurons with different response behaviors. Information curves as in A. Spike trains are split by current force direction and colored by previous force direction. Examples are of a neuron with mixed tuning (left), a neuron that signals information about the previous stimulus (middle), and a neuron that signals information about the current one (right). (**C**) Proportion of FA-1 (blue), SA-1 (green), and SA-2 (purple) neurons showing different response behaviors during the protraction phase. Most neurons signaled the current force direction, but around 35% either signaled the previous force direction or showed mixed response behavior. (**D**) Information transmitted about force direction for neurons tuned to the current force direction for the regular and the irregular sequence. Information decreases considerably for the irregular sequence, even in neurons responding strongly to the current direction.

We also assessed information about the preceding force direction contained in the neural responses during the protraction phase in the irregular sequence. We reasoned that if the preceding loading systematically affected contactor position in the subsequent trial, then neurons might carry information about past stimulation in their responses. Consistent with this idea, such information was present, albeit at a relatively low level. Averaged across all neurons of each type, information about the preceding force direction increased for about 60–70ms into the protraction phase, after which it appeared to plateau or decrease (yellow traces in *Figure 6A*; p < 0.001 for each type, Wilcoxon one-sample signed rank tests based on information values halfway through the protraction phase compared against zero information). Irrespective of neuron type, signaling of the preceding as well as the current force direction could vary substantially between neurons, with some carrying information mostly about the current direction, and others about the preceding one (see example information traces for individual neurons in *Figure 6B*). We quantified the diversity amongst neurons in this respect based on whether they primarily conveyed information about the present force direction, the preceding force direction, or a mix of both (including 53 out of 67 FA-1, 67 out of 73 SA-1 and 35 out of 41 SA-2; see Materials and methods). Most neurons primarily signaled information about the current force direction (46 SA-1, 35 FA-1, and 18 SA-2 neurons, see *Figure 6C*). Fewer showed mixed tuning, and those that did signaled preceding force direction early during the protraction phase and then switched to information about the current loading direction (16 SA-1, 9 FA-1, and 13 SA-2). Finally, some neurons (9 FA-1, 4 SA-1 and 4 SA-2) primarily signaled information about preceding force direction. We found no significant differences in the relative frequency of FA-1, SA-1, and SA-2 neurons that responded to the current or previous stimulation, or both ($\chi^2(3, 155) = 6.95$, $p = 0.14$).

Notably, information transmission in both FA-1 and SA-1 neurons decreased between the regular and irregular sequence, even when only considering those neurons that predominantly signaled information about the current force direction (*Figure 6D*), confirming that information transmission about the ongoing stimulus was still affected by the fingertip's viscoelastic memory (p < 0.001 for both types, p = 0.18 for SA-2, paired Wilcoxon signed rank tests).

We found no straightforward relationship between a neuron's sensitivity to current and previous stimulation and its termination site in fingertip skin. Specifically, there was no statistically significant effect of the distance between a neuron's receptive field center and the primary contact site of the stimulus surface on whether neurons signaled current, prior, or mixed information for SA-1 (Kruskal-Wallis test, $H(2) = 3.86$, $p = 0.15$) or SA-2 neurons ($H(2) = 0.75$, $p = 0.69$). However, a significant difference emerged for FA-1 neurons ($H(2) = 8.66$, $p = 0.01$), indicating that neurons terminating closer to the stimulation site on the flat part of the fingertip were more likely to signal past or mixed information.

Overall, our findings suggest that past loading reduces the information conveyed by these neurons about the direction of prevailing fingertip forces. Nevertheless, neurons of all three types retain information about the previous force direction, albeit with substantial variability among individual neurons. This variability may be partially attributed to the specific locations on the fingertip where neurons terminate, determining the local strain patterns they experience. Additionally, differences in the specific aspects of local strain to which individual neurons respond could further contribute to this heterogeneity.

## Neural responses in the interstimulus period

SA-2 neurons can exhibit ongoing activity without external stimulation and sense tension states in collagenous fiber strands in dermal and subdermal tissues (*Knibestöl, 1975*; *Birznieks et al., 2009*; *Johansson, 1978*; *Chambers et al., 1972*). Since the deformation of the fingertip by the force stimuli was mostly absorbed by such tissues, we hypothesized that SA-2 neurons active during the interstimulus periods might convey ongoing information about the viscoelastic state of the fingertip during the recovery from the recurrently applied loadings. A subset of SA-2 neurons (20 out of 41) exhibited such activity (see *Figure 7A* for three examples). Calculation of mutual information indicated that some individual neurons were highly informative about the preceding stimulus direction, but the time of maximal information transmission could occur at different points (see examples in *Figure 7B*). On average, SA-2 neurons provided low but continuous information about the preceding force direction throughout the interstimulus period, which was highest at the start of the interstimulus period and tended to decrease slightly over time (*Figure 7C*, yellow trace). This decrease was likely driven by the gradual relaxation of the fingertip.

We next asked whether SA-2 neural activity at the population level could track the deformation state of the fingertip during the interstimulus period. We calculated a low-dimensional representation of the SA-2 population activity to compare with the fingertip deformation at three different time points: in the middle and at the end of the interstimulus period, and again at the end of the protraction phase. Specifically, based on activity recorded during the irregular sequence, we calculated pairwise spike distances across all 25 trials (5 force directions ×5 trials) for each neuron, providing a measure for how distinct the activity of this neuron was across different trials. This process yielded a matrix, which was averaged across all active SA-2 neurons. We then used multi-dimensional scaling to place each of the 25 types of trials into a two-dimensional space. Using Procrustes analysis, finally, we rotated and scaled the responses to match the recorded tangential contactor positions, which were calculated for the same 25 types of trial at the corresponding time points and averaged across fingertips (*Figure 8A*). Notably, there was a good match between the neural representation and the fingertip deformation. For both measures, different trial types were clustered according to the force direction of the preceding loading throughout the interstimulus period, but cluster separation decreased as time progressed. Then trial types diverged during the protraction phase, and at the end of the phase, they were clustered according to the current force direction. To quantify the similarity of the two representations, for each we calculated two measures. The 'total variance' across all 25 trial types indicates the general level of variability (see black dashed ellipse in the top panel of *Figure 8A*). The 'direction variance', which was calculated over trials representing the same preceding force direction and then averaged, indicates variability within clusters of trials with the same preceding direction (see dashed orange ellipse in the top panel of *Figure 8A*). Skin positions and SA-2 population activity displayed

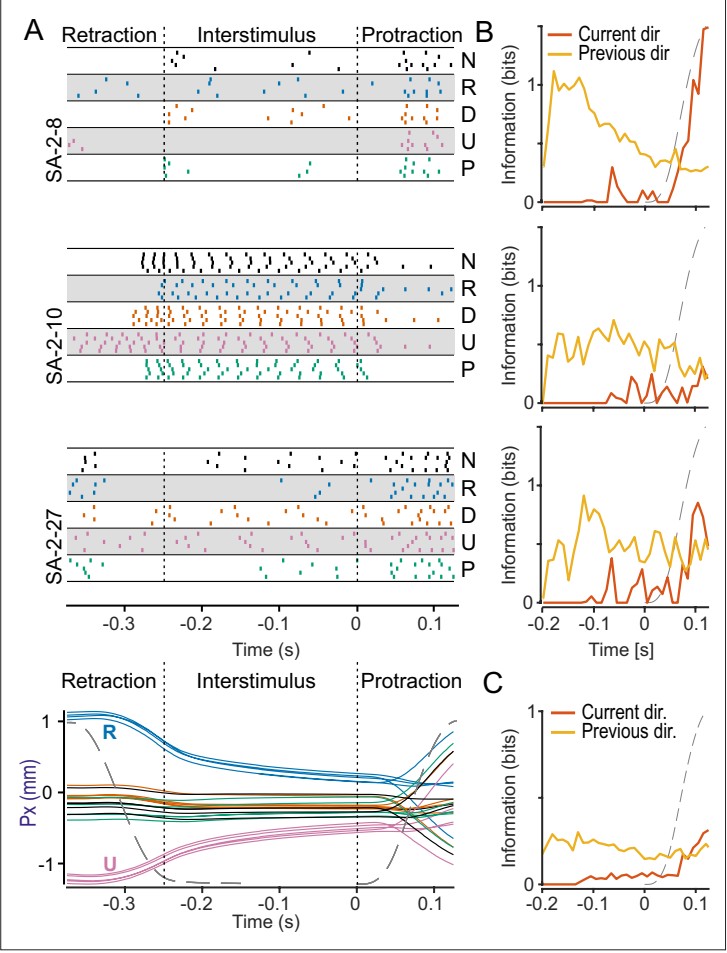

**Figure 7.** SA-2 neurons signal information about previous forces in the interstimulus period. (**A**) Spike raster plots for three SA-2 neurons recorded during the interstimulus periods of the irregular sequence. Each dot represents an action potential, and the colors indicate the preceding force direction. Note that the responses differ systematically based on the previous force direction. Bottom panel: To illustrate the effect of previous force direction on the deformation state of the fingertip during the interstimulus period, the average contactor position in the ulnar-radial (U–R) direction during the corresponding irregular stimulation sequences is shown (on the same time scale as upper panels). Colors indicate the force direction in the preceding trial corresponding to the color coding in A. (**B**) Average mutual information about force direction in the test trial (orange line) or the preceding trial (yellow line) during the interstimulus period and during the subsequent protraction phase for the same three neurons shown in panel A. (**C**) Average mutual information across all SA-2 neurons active in the interstimulus period.

a similar variance pattern (*Figure 8B*). During the interstimulus period, the total variance was much larger than the direction variance, signifying a marked clustering according to the preceding force direction, but this difference tended to decrease with time. In contrast, both variances were large and roughly equal at the end of the protraction phase, indicating little clustering based on the preceding force direction. Taken together, these results suggest that SA-2 neurons can continuously signal the mechanical state of the fingertip even in the absence of externally applied stimulation.

## Discussion

We found that the viscoelasticity of the fingertip affects signals in first-order tactile neurons when responding to fingertip loadings mimicking those experienced in everyday object manipulation tasks. Such tasks involve applying forces of different magnitudes and directions in rapid succession, such as during grasping and transporting objects, handicraft, cooking, cleaning, and food gathering. The

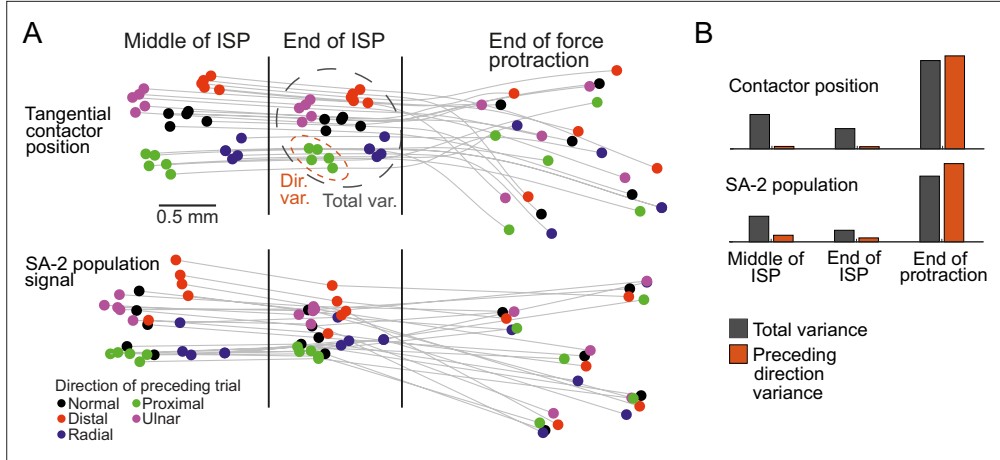

**Figure 8.** Continuous representation of fingertip viscoelastic state in the SA-2 population. (**A**) Colored dots indicate tangential contactor positions (top) and their representation in the SA-2 population signal (bottom) at three different times: the middle of the interstimulus period (–0.125 s), the end of the interstimulus period (0 s), and the end of the subsequent protraction phase (0.125 s). The colors of the markers denote the force direction in the directly preceding trial. Contactor position is the two-dimensional position in the tangential plane. SA-2 representations are derived from average spike train distances across the different trials, visualized in a two-dimensional space using multidimensional scaling and aligned with the contactor positions using Procrustes analysis (see Methods for details). (**B**) Averaged total variance and within-direction variance for contactor position and SA-2 population signal representations at the same three time points as in A. As illustrated by the dashed ellipses in A, total variance denotes the two-dimensional variance across all trials, while within-direction variance denotes the variance for trials belonging to the same preceding force direction. Higher total than within-direction variance indicates that data for trials in the same preceding direction are more clustered than data for trials in all preceding directions, which is required for discrimination of preceding force direction.

neurons' signaling of force direction was significantly influenced by the direction of the preceding loading, even if it varied by only 20 degrees relative to the perpendicular direction. This effect was most prominent during the force protraction phase, which is when neurons typically signal the most information about force direction. However, there was heterogeneity in how individual neurons behaved. Some neurons primarily signaled information about the current direction, while others signaled both the current and previous direction, or even primarily the preceding direction. Our results also indicate that neurons can signal information related to the fingertip's viscoelastic deformation state even between loadings: SA-2 neurons' ongoing impulse activity between loadings was influenced by the previous loading direction. This diversity suggests that, at the population level, first-order neurons carry information about the fingertip's current viscoelastic state, which within it contains a memory of past stimulation.

## Origins of neural heterogeneity

The observed heterogeneity between neurons of the same type is expected, given that their transduction sites were distributed widely within the fingertip skin and they are sensitive to the local stresses and strains at their transduction site, rather than the global deformation of the fingertip (*Birznieks et al., 2001*; *Saal et al., 2009*). Thus, the viscoelastic memory of the preceding loading would have modulated the pattern of strain changes in the fingertip differently depending on where their receptor organs are situated in the fingertip.

In addition to the receptor organ locations, the variation in sensitivity among neurons to fingertip deformations in response to both previous and current loadings would stem from the fingertip's geometry and its complex composite material properties. Possible inherent directional preferences of the receptor organs, attributed to their microanatomy, could also be significant. However, mechanical anisotropy, particularly within the viscoelastic subcutaneous tissue of the fingertip induced by intricately oriented collagen fiber strands forming fat columns in the pulp (*Hauck et al., 2004*), is likely to play a crucial role. This anisotropy would shape the dynamic pattern of strain changes at neurons' receptor sites, intricately influencing a neuron's sensitivity not only to current but also to preceding

loadings. Indeed, recent modeling efforts suggest that such mechanical anisotropy strongly influences the spatiotemporal distribution of stresses and strains across the fingertip (*Duprez et al., 2024*). The resulting diversity in the sensitivities of neurons might enhance the overall information collected and relayed to the brain by the neuronal population, facilitating the discrimination between tactile stimuli or mechanical states of the fingertip (see *Rongala et al., 2024*; *Corniani et al., 2022*; *Tummala et al., 2023*, for more extensive explorations of this idea).

Our regression analysis, which examined how well different aspects of contactor behavior could predict the time-varying firing rates of individual neurons, highlights the complexity of neural responses to fingertip mechanics. This conclusion is supported by three key findings. First, models incorporating interactions between contactor position and velocity outperformed simpler models. Second, the importance of independent variables varied among neurons of the same type, reflecting differences in sensitivity to contactor position, velocity, and their interactions. Third, prediction accuracy varied significantly across neurons of the same type. The limited accuracy of the two-way interaction model in replicating detailed firing rate profiles of individual neurons underscores the intricate relationship between fingertip events and neural activity. Although a three-way interaction model was evaluated, it provided only marginal improvements in variance explained (data not shown).

Predictors that showed consistently high importance across neurons were position-velocity interactions, revealed by the two-way interaction model. This finding suggests that neuronal responses to fingertip deformation changes (reflected by contactor velocity) are significantly influenced by the fingertip's current deformation state (indicated by contactor position), modulated by previous fingertip loading through viscoelastic effects. Direct observations support this conclusion. First, variation in neuronal response to force protraction, depending on previous loading direction, paralleled changes in contactor position and path, whereas contactor velocity, the primary driver of the neuronal response, remained generally aligned with the force direction (*Figure 2C–E*). Second, FA-1 neurons, which displayed minimal sensitivity to contactor position alone, exhibited greatly improved prediction accuracy when position and velocity variables interacted, demonstrating that the fingertip's deformation state modulates the neurons' dynamic responses.

## Differences between neuron types in sensitivity to viscoelasticity

We also observed some differences between neuron types regarding how variation in the preceding force direction affected their signaling during the current loading. The impact of stimulation history during the force protraction phase was more pronounced in FA-1 and SA-1, compared to SA-2 neurons. It is possible that the neurons' response properties accounted for this difference: type I neurons are primarily sensitive to deformation of the fingerprint ridges (*Sukumar et al., 2022*; *Jarocka et al., 2021*), while SA-2 neurons primarily signal tension states in deeper dermal and subdermal tissues (*Knibestöl, 1975*; *Birznieks et al., 2009*; *Johansson, 1978*; *Chambers et al., 1972*). The fingertip deformation changes during the loadings can be seen as twofold. First, the force applied by the contactor induced bulk deformation changes of the fingertip that depended on the force direction. As pressure increases in the pulp, the pulp tissue bulges at the end and sides of the fingertip. Simultaneously, the tangential force component amplifies the bulging in the direction of the force while stretching the skin on the opposite side. This effect is attributed to the fact that the friction between the stimulation surface and the skin was high enough to prevent it from sliding over the fingertip. Bulk deformation changes are closely linked to widespread alterations in stress and strain distribution in deeper tissues, making the SA-2 neuron population the preferred signaling source for such changes. For example, it has been demonstrated that experimentally induced compliance changes of the finger pulp using venous occlusion readily influenced activity of SA-2, but not SA-1 neurons (*Hudson et al., 2015*). However, concurrent with the bulk deformation changes, the fingertip skin would undergo direction-dependent surface deformation changes involving planar tensile strain changes and partial slippage peripherally within the contact surface (*Delhaye et al., 2014*; *Delhaye et al., 2016*; *Willemet et al., 2021*). Changes in planar tensile strain per se might excite neurons of either type, given that even neurons terminating outside the contact area can respond to fingertip loadings (*Bisley et al., 2000*; *Birznieks et al., 2001*). However, partial slippages occurring during fingertip loadings, where some parts of the fingertip-object interface slip while others remain stuck, excite SA-1 and especially FA-1 neurons most intensely (*Johansson and Westling, 1987*; *Srinivasan et al., 1990*; *Khamis et al., 2014*; *Delhaye et al., 2021*). A substantial part of the sensitivity of FA-1 and SA-1 neurons

to stimulation history could therefore be attributed to the preceding force direction affecting the location and timing of partial slips. That is, due to the viscoelastic memory of the fingertip, previous loadings in different directions could result in different patterns of planar tensile stress changes under a given loading condition, which would affect how and where the local partial slips occur. The responsiveness of type I neurons to partial slippage might also explain their apparently higher sensitivity to stimulation history compared to SA-2s. In sum, we believe that both bulk and superficial deformation changes play a role in the activation of the tactile neurons during the fingertip loadings.

Our regression analysis of contactor behavior and time-varying firing rates corroborates that the different neuron types exhibit varying sensitivities to mechanical fingertip events during fingertip loadings. The observation that predictions were most accurate for SA-2 neurons aligns with the understanding that contactor behavior during loading primarily reflects bulk deformation changes, best signaled by SA-2 neurons. The lower prediction accuracy for SA-1 and, particularly, FA-1 neurons suggests that these are influenced by additional factors that are less effectively or not represented in the behavior of the contactor, specifically, direction-dependent planar tensile strain changes in the skin and partial slippages, which likely predominantly affect FA-1 neurons (see e.g. *Delhaye et al., 2021*).

## Continuous signaling of viscoelastic state by SA-2 neurons

That SA-2 neurons signaled the viscoelastic state of the fingertip in periods between loadings is consistent with previous ideas that SA-2 neurons continuously measure stresses in collagen fiber strands that run within and between dermal and subdermal tissues (*Vallbo and Johansson, 1984*; *Chambers et al., 1972*; *Birznieks et al., 2009*). Indeed, their relatively low dynamic sensitivity to externally applied loads and the well-sustained response to maintained loadings suggest that SA-2 neurons are tailored to encode slow viscoelastic and quasistatic events occurring in dermal and subdermal tissues (*Westling and Johansson, 1987*; *Birznieks et al., 2001*; *Birznieks et al., 2009*). Furthermore, even in the absence of externally applied stimulation, they can exhibit ongoing impulsive activity, which suggests they are capable of monitoring inherent mechanical tension patterns in dermal and subdermal tissues in the skin in the unloaded state (*Knibestöl, 1975*; *Johansson, 1978*). In particular, changes in the tension patterns via finger and hand movement without external stimulation can also modulate this activity. We believe that by constantly transmitting information related to tissues' viscoelastic state, SA-2 neurons could help keep the brain updated about the current mechanical state of body parts. In agreement with this view, peripheral nerve blocks affect the perceived image of body parts such as the arm and fingers (*Inui et al., 2011*; *Walsh et al., 2015*; *Melzack and Bromage, 1973*). Likewise, although sensations elicited by electrical stimulation of single SA-2 neurons innervating the hand have been elusive (*Kunesch et al., 1995*; *Ochoa and Torebjörk, 1983*), a recent study indicates that they can give rise to distinct sensations that include the experience of diffuse skin deformation (*Watkins et al., 2022*). By continuously informing about the viscoelastic state of the fingertips, SA-2 neurons could help the brain in computing accurate motor commands for interactions with objects in manipulation and haptic tasks by updating reference frames for interpreting information signaled by type I neurons.

## Consequences for haptic performance and perception

Although tactile information in general is crucial for planning and executing motor actions in such tasks, to our knowledge, a possible influence of fingertip viscoelasticity on task performance has not been systematically investigated. Similarly, there is a lack of systematic investigation of potential effects of fingertip viscoelasticity on performance in tactile psychophysical tasks conducted during passive touch. Therefore, it is unclear whether viscoelasticity limits performance or if it is compensated for in some way. Our findings indicate that the population of tactile neurons that innervate a fingertip encode continuous information about the fingertip's viscoelastic deformation state. This information could potentially aid the brain in managing the effects of viscoelasticity on tactile coding and fingertip actions. For instance, the brain could intermittently use this information to estimate the state of the fingertip during planning and evaluation of tactile-based actions (*Johansson and Flanagan, 2009*). It is also conceivable that the brain continuously represents the current deformation state of the fingertips using online population information. However, such processing might require considerable computational resources. In any case, the viscoelastic deformability of the fingertips plays a pivotal

role in supporting the diverse functions of the fingers. For example, it allows for cushioned contact with objects featuring hard surfaces and allows the skin to conform to object shapes, enabling the extraction of tactile information about objects' 3D shapes and fine surface properties. Moreover, deformability is essential for the effective grasping and manipulation of objects. This is achieved, among other benefits, by expanding the contact surface, thereby reducing local pressure on the skin under stronger forces and enabling tactile signaling of friction conditions within the contact surface for control of grasp stability. Throughout, continuous acquisition of information about various aspects of the current state of the fingertip and its skin by tactile neurons is essential for the functional inter-action between the brain and the fingers. In light of this, the viscoelastic memory effect on tactile signaling of fingertip forces can be perceived as a by-product of an overall optimization process within prevailing biological constraints.

## Materials and methods
### General procedure and study participants

The study was conducted in accordance with the Declaration of Helsinki, apart from pre-registration in a database. Written informed consent was obtained from all participants prior to their inclusion in the study, including consent for publication of anonymized data. A total of 33 healthy adults (21 females and 12 males, aged 19–30 years) participated voluntarily after being fully informed about the study procedures. The study protocol was reviewed and approved by the Research Ethics Committee at Umeå University (Item 175/02, registration number 02–148). The general experimental methodology, procedure, and apparatus have been described previously (*Birznieks et al., 2001*), as well as other aspects of the same experimental data than those analyzed here (*Birznieks et al., 2001*; *Jenmalm et al., 2003*; *Saal et al., 2009*; *Johansson and Birznieks, 2004*). Briefly, action potentials (spikes) in axons of single first-order tactile neurons that terminated in the distal segment of the index, middle or ring finger were recorded with tungsten needle electrodes inserted into the median nerve at the level of the upper arm 0.5–0.6 m from the fingertips (*Vallbo and Hagbarth, 1968*). For neurons with cutaneous receptive fields on the distal segment of a finger, force stimuli were applied to its fingertip in five different directions by means of a custom-built robot. The fingertip was stabilized by gluing the nail to a firmly fixed metal plate. Force was transferred through a circular plane (30 mm diameter) that was centered on the midpoint of a line extending in the proximal-distal direction from the papillary whorl to the distal end of the finger. The stimulating surface was oriented parallel to the skin at this primary contact site (see *Figure 1B*), which was located approximately at the center of the flat part of the fingertip's volar surface and serves as a primary target for object contact in fine manipulation tasks engaging 'tip-to-tip' precision grips (*Christel et al., 1998*). To ensure that friction between the contactor and the skin was sufficiently high to prevent slips, the surface was coated with silicon carbide grains (50–100 μm), approximating the finish of smooth sandpaper.

### Force stimuli
#### Force parameters

One of the five directions of force stimulation was normal (N) to the skin surface at the primary contact site, and the other four were angled 20° to the normal direction in the radial (R), distal (D), ulnar (U), and proximal (P) directions, respectively. All force stimuli were superimposed on 0.2 N background force normal to the skin and consisted of a force protraction phase (125ms), a plateau phase at 4 N normal force (250ms), and a force retraction phase (125ms) (*Figure 1C*). In the four trials with a tangential force component, the tangential force was 1.4 N at the force plateau. The time course of the force changes followed a half-sinusoid (sine wave frequency of 4 Hz). The position and orientation of the stimulation surface in relation to the primary contact site were maintained when the loading contained tangential force components. That is, the friction between the stimulation surface and the skin was high enough to prevent frictional slips. The interval between successive fingertip loadings was 250ms. Thus, the frequency of recurrent fingertip loadings (1.3 Hz) was representative of the frequency at which tactile events follow each other during dexterous object manipulation tasks (see e.g. *Draper, 1994*; *Kunesch et al., 1989*; *Teulings and Maarse, 1984*). Likewise, in trials with a tangential force component, the magnitudes, directions, and time courses of the fingertip forces were similar to those

employed when people use a precision grip to lift an object weighing 250–300 g (*Johansson and Westling, 1984*; *Westling and Johansson, 1984*).

## Stimulation sequences

Two force stimulation sequences containing trials in each of the five different directions were delivered repeatedly. In the regular sequence, the trial order was the same (R, D, U, P, N) for all repetitions of the sequence. The regular sequence was presented six times, but to standardize the stimulation history for all trials, only data obtained during the last five repetitions was included in the analysis (5 force directions ×5 trials). Immediately after the completion of the regular sequence, the irregular sequence was delivered in which the trial order was systematically changed over repeats of the sequence. Each of the five loading directions (R, D, U, P, and N) was presented five times in such a way that each loading was preceded once by loading in each of the five directions (5 force directions ×5 trials).

## Neural sample

The neurons recorded from were classified as fast-adapting type I (FA-1), fast-adapting type II (FA-2), slowly-adapting type I (SA-1), and slowly-adapting type II (SA-2) according to criteria described previously (*Johansson and Vallbo, 1983*; *Vallbo and Johansson, 1984*). Briefly, FA afferents respond only to changes in skin deformation, whereas SA afferents show an ongoing response during periods of static skin deformation. Type I afferents (FA-1 and SA-1) possess small and well-delineated receptive fields if probed by light, pointed skin indentations, while the receptive fields of type II afferents (FA-2 and SA-2) are often large and poorly defined (see *Vallbo and Johansson, 1984*, for further details). The present analysis included 60 FA-1, 73 SA-1, and 41 SA-2 neurons terminating in the glabrous skin of the terminal segment of digits II, III, or IV. Other functional aspects of these neurons have been explored in previous studies (*Birznieks et al., 2001*; *Jenmalm et al., 2003*; *Johansson and Birznieks, 2004*; *Saal et al., 2009*). To ensure a reasonably balanced representation of the three neuron types in the sample, given the lower density of slowly adapting neuron types in fingertip compared with FA-1 neurons (*Johansson and Vallbo, 1979*), slowly adapting neuron types were intentionally prioritized during prerecording search for unitary action potentials. Initially, 10 FA-2 neurons were also included in the analysis. But their responsiveness during the experiment was remarkably low, and unlike the other neuron types, their responses were rarely affected by force stimuli. Specifically, only one of the observed FA-2 neurons responded during the force protraction phases. Due to the lack of clear stimulus-driven responses, FA-2 neurons were subsequently excluded from further analysis.

## Analysis

### Quantifying mechanical and neural response variability

We assessed the variability in displacements and velocities by calculating the standard deviation of the displacement and velocity signals at each time point over the two-dimensional tangential plane across repeated trials of the same force direction, using Euclidean distance as the metric. Variability was assessed separately for trials in the regular and the irregular sequence and averaged across all fingertips recorded from. Similarly, when assessing the variability of the neural responses, we calculated the standard deviation of the instantaneous firing rate across trials in the same force direction over time. Instantaneous firing rate was calculated as the inverse of the interspike interval at a given time point. Again, this analysis was run separately on data from the regular and the irregular sequence. The resulting standard deviation traces were first averaged over different force directions and then over all neurons from the same class.

### Predicting time-varying firing rates from contactor movements

This analysis was conducted in Python (v3.13) with *pandas* for data handling, *numpy* for numerical operations, and *scikit-learn* for model fitting and evaluation.

To assess how well individual neurons' time-varying firing rates could be predicted from simultaneous contactor movements, we fitted multiple linear regression models (see *Khamis et al., 2015*, for a similar approach). This analysis focused on the force protraction phase of the irregular sequence, where neurons were most responsive and sensitive to stimulation history. Data from 100ms before to 100ms after the protraction phase (between –0.100 s and 0.225 s relative to protraction onset) were included for each trial. Neurons were included if they fired at least two action potentials during the

force protraction phase and the following 100ms in at least five of the 25 trials. This ensured sufficient variability in firing rates for meaningful regression analysis, resulting in 68 SA-1, 38 SA-2, and 51 FA-1 neurons being included.

Contractor position signals digitized at 400 Hz were linearly interpolated to 1000 Hz. Instantaneous firing rates, derived from action potentials sampled at 12.8 kHz, were resampled at 1000 Hz to align with position signals. A Gaussian filter ($\sigma = 10$ ms, cutoff $\sim 16$ Hz) was applied to the firing rate as well as to the position signals before differentiation. To account for axonal conduction (8–15ms) and sensory transduction delays (1–5ms), firing rates were advanced by 15ms to align approximately with independent variables.

Regressions were performed using scikit-learn's *Ridge* and *RidgeCV* regressors, which apply L2 regularization to mitigate overfitting. Hyperparameter tuning for the regularization parameter (alpha) was performed using *GridSearchCV* with a predefined range (0.001–1000.0), incorporating five-fold cross-validation to select the best value. To minimize overfitting risks, model performance was further validated with independent five-fold cross-validation (*KFold*), and $R^2$ scores were computed using *cross_val_score*.

We constructed four linear regression models with increasing complexity: (1) Position-only, using three-dimensional contactor positions (Px, Py, Pz); (2) Velocity-only, using three-dimensional velocities (dPx, dPy, dPz); (3) Combined, including all position and velocity signals (6 predictors); and (4) Interaction, including all signals and their two-way interactions (21 predictors). All features were standardized using *StandardScaler* to improve regularization and model convergence. *PolynomialFeatures* generated second-order interaction terms for the interaction model. Feature importance was evaluated with *permutation_importance*, and simpler models were built using the most important features. These models were validated through cross-validation to assess retained explanatory power.

## Calculation of information transmission

To assess the amount of information about force direction conveyed in responses of individual neurons, for each neuron we calculated a lower bound on the information transmitted about both the current and the previous force direction. We used metric space analysis (*Victor and Purpura, 1997*), which employs a classifier on the neural response data to distinguish different force directions, calculates a confusion matrix from several runs of the classifier, and finally computes a lower bound on the mutual information from the confusion matrix. Details of the specific implementation used in the present study have been described previously (*Saal et al., 2009*). In short, we first sorted the trial data either by the force direction in the current trial (to assess coding of current force direction) or by the force direction in the previous trial (to assess whether neurons responded to the viscoelastic 'memory' of the previous trial). We then computed spike distances between all pairs of spike trains according to a spike-timing based distance metric that assesses the 'cost' for transforming the first spike train in the pair into the second, by adding or removing individual spikes, or shifting existing ones in time. For analyses of the protraction phase, this spike distance was computed at a temporal resolution of 8ms, as this was determined in a previous study to be close to the optimum for maximal information transmission for the present experimental data (*Saal et al., 2009*). For the analysis of SA-2 responses during the interstimulus period, we used a lower temporal resolution of 32ms instead, to adjust for the lower firing rates of these neurons in the absence of externally applied stimulation. Each individual trial was then classified as originating from the force direction to which its average distance was lowest. In this way, a confusion matrix was generated for each neuron, with each entry denoting the number of times a neural response from a given force direction was classified as having originated from that direction or another one. Finally, we used the confusion matrix to compute a lower bound of the mutual information. To quantify the bias of this estimate, we reassigned neural responses from individual trials to random conditions and recalculated the mutual information. The bias term was set as the average of the outcome of 10 such random assignments and subtracted from the estimate of the mutual information. This analysis was run on successively longer time windows starting at stimulus onset (when calculating information during the protraction phase) or at the start of the interstimulus period (when calculating information for this period), respectively, and extending until the end of the considered time window in 5ms increments. In this way, it could be assessed how neural coding of force direction evolved over time. The whole analysis was run twice on the irregular sequence data: once with stimuli grouped according to the current force direction (to assess the information

conveyed about the stimulus), and a second time with stimuli grouped according to the preceding trial (to assess information conveyed in the neural responses about the past stimulus). The analysis was also run on the regular sequence; because effects due to previous and current stimulation cannot be distinguished in this data set due to the fixed trial order, this analysis yielded a single information value that represents the total amount of information available when past loading is held constant.

For the analysis covering the protraction phase, we also assigned each neuron to one of three groups, based on their information curves: those that primarily conveyed information about the current force direction ('current'), those that conveyed information about the preceding force direction ('previous'), and those that conveyed different types of information at different times during the protraction phase ('mixed'). To do this, we first excluded neurons that responded too weakly or erratically, as assessed by whether they conveyed information (above 0) about either past or present force direction for at least 10 different time windows during the protraction phase. This left 53 FA-1, 67 SA-1, and 35 SA-2 neurons. For each time window containing non-zero information, we then measured whether more information was conveyed about the present or the past force direction. Neurons that conveyed information about the current force direction in 70% or more of time windows were classed as 'current'; neurons with 30% or less were classed as 'previous'; and all other neurons were classed as 'mixed'.

## Analysis of SA-2 population responses

To derive a representation of the SA-2 population response, we took the spike train distances (see above) calculated at three different time points (in the middle and at the end of the interstimulus period at –0.125 s and 0 s relative to protraction onset, and at the end of the protraction phase at 0.125 s) for the irregular sequence and summed these distances across all 20 SA-2 neurons that were tonically active during the interstimulus period. This yielded a 25-by-25 matrix, with each entry denoting how dissimilar a given trial in the irregular sequence was from another one on the population level. We then employed multidimensional scaling, which aims to embed data points in a high-dimensional space such that their Euclidean distances adhere to those in the original distance matrix as closely as possible. We retained the first two dimensions of this space, resulting in each trial now occupying a position in a two-dimensional space. Finally, using Procrustes analysis, we rotated and scaled this representation to match it to the two-dimensional skin deformations in the plane tangential to the fingertip surface at the three different time points. The main idea behind this analysis is that if time-varying skin deformations are encoded at the SA-2 population level, then more dissimilar skin deformations should lead to more dissimilar neural responses. The analysis tests the extent to which this idea is true.

## Statistics

As mechanical and neural measures were generally not normally distributed, we used non-parametric tests for all statistical analyses on these variables. Specifically, for all analyses comparing the regular and irregular sequences, we used paired Wilcoxon signed rank tests. When making multiple comparisons within the same analysis, we used Bonferroni corrections, in which case the resulting p values are reported as $p_{corrected}$.

To assess the effect of regression model and neuron type on $R^2$ scores, a two-way mixed-design ANOVA was conducted. The regression model (1–4) served as the within-group factor, and neuron type (FA-1, SA-1, SA-2) as the between-group factor. Correlation coefficients were Fisher-transformed into Z-scores for parametric testing and averaging, then converted back to correlation coefficients for reporting. Results are expressed as coefficients of determination ($R^2$). For post hoc comparisons, Tukey's HSD test was used to control the family-wise error rate.

# Additional information

## Funding

| Funder | Grant reference number | Author |
|---|---|---|
| Leverhulme Trust | RPG-2022-031 | Hannes P Saal |
| Vetenskapsrådet | 08667 | Roland S Johansson |

The funders had no role in study design, data collection and interpretation, or the decision to submit the work for publication.

## Author contributions

Hannes P Saal, Conceptualization, Software, Visualization, Methodology, Writing – original draft, Writing – review and editing; Ingvars Birznieks, Conceptualization, Investigation, Methodology, Writing – review and editing; Roland S Johansson, Conceptualization, Data curation, Software, Supervision, Investigation, Visualization, Methodology, Writing – original draft, Writing – review and editing

## Author ORCIDs

Hannes P Saal ⓘ https://orcid.org/0000-0002-7544-0196
Ingvars Birznieks ⓘ https://orcid.org/0000-0003-4916-1254
Roland S Johansson ⓘ https://orcid.org/0000-0003-3288-8326

## Ethics

The study was conducted in accordance with the Declaration of Helsinki, apart from pre-registration in a database. Written informed consent was obtained from all participants prior to their inclusion in the study, including consent for publication of anonymized data. A total of 33 healthy adults (21 females and 12 males, aged 19-30 years) participated voluntarily after being fully informed about the study procedures. The study protocol was reviewed and approved by the Research Ethics Committee at Umeå University (Item 175/02, registration number 02-148).

Reviewer #1 (Public review): https://doi.org/10.7554/eLife.89616.3.sa1
Reviewer #2 (Public review): https://doi.org/10.7554/eLife.89616.3.sa2
Author response https://doi.org/10.7554/eLife.89616.3.sa3

# Additional files

## Supplementary files

MDAR checklist

## Data availability

All data has been uploaded to ORDA, the University of Sheffield data repository, and is available at https://doi.org/10.15131/shef.data.23054657.

The following dataset was generated:

| Author(s) | Year | Dataset title | Dataset URL | Database and Identifier |
|---|---|---|---|---|
| Saal HP, Birznieks I, Johansson RS | 2023 | Memory at your fingertips: how viscoelasticity affects tactile neuron signaling | https://doi.org/10.15131/shef.data.23054657 | ORDA Online Research Data, 10.15131/shef.data.23054657 |

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
