## [Editor Report · eLife Assessment]

The **fundamental** findings reported here provide insight into how the viscoelasticity of the fingertip skin influences the activity of mechanoreceptive afferents and thus the neural coding of force in humans. The basic principle studied was whether and to what extent the previous applied force directions impact the firing of FA-1, SA-1 and SA-2 neurons during the current applied force directions. The data and analyses are **compelling** and will be helpful for modeling the neural representations of force in the context of object grasping and manipulation.

---

## [Referee Report · Reviewer #1 (Public review)]

The authors investigate how the viscoelasticity of the fingertip skin can affect the firing of mechanoreceptive afferents and they find a clear effect of recent physical skin state (memory), which is different between afferents. The manuscript is extremely well-written and well-presented. It uses a large dataset of low threshold mechanoreceptive afferents in the fingertip, where it is particularly noteworthy that the SA-2s have been thoroughly analyzed and play an important role here. They point out in the introduction the importance of the non-linear dynamics of the event when an external stimulus contacts the skin, to the point at which this information is picked up by receptors. Although clearly correlated, these are different processes, and it has been very well-explained throughout. I have some comments and ideas that the authors could think about that could further improve their already very interesting paper. Overall, the authors have more than achieved their aims, where their results very much support the conclusions and provoke many further questions. This impact of the previous dynamics of skin affecting current state can be explored further in so many ways and may help us in understanding skin aging and the effects of anatomical changes of the skin better.

Comments on revised submission:

The authors have taken all my considerations into account and provided excellent responses to them. They have modified their paper accordingly, which improves its clarity even more. Very interesting work and I have no further comments.

---

## [Referee Report · Reviewer #2 (Public review)]

Summary:

The authors sought to identify the impact skin viscoelasticity has on neural signalling of contact forces that are representative of those experienced during normal tactile behaviour. The evidence presented in the analyses indicate there is a clear effect of viscoelasticity on the imposed skin movements from a force-controlled stimulus. Both skin mechanics and evoked afferent firing were affected based on prior stimulation, which has not previously been thoroughly explored. This study outlines that viscoelastic effects have an important impact on encoding in the tactile system, which should be considered in the design and interpretation of future studies. Viscoelasticity was shown to affect the mechanical skin deflections and stresses/strains imposed by previous and current interaction force, and also the resultant neuronal signalling. The result of this was an impaired coding of contact forces based upon previous stimulation. The authors may be able to strengthen their findings, by using the existing data to further explore the link between skin mechanics and neural signalling, giving a clearer picture than demonstrating shared variability. This is not a critical addition, but I believe would strengthen the work and make it more generally applicable.

Strengths:

-Elegant design of the study. Direct measurements have been made from the tactile sensory neurons to give detailed information on touch encoding. Experiments have been well designed and the forces/displacements have been thoroughly controlled and measured to give accurate measurements of global skin mechanics during a set of controlled mechanical stimuli.

-Analytical techniques used. Analysis of fundamental information coding and information representation in the sensory afferents reveals dynamic coding properties to develop putative models of the neural representation of force. This advanced analysis method has been applied to a large dataset to study neural encoding of force, the temporal dynamics of this, and the variability in this.

Weaknesses:

-Lack of exploration of the variation in neural responses. Although there is a viscoelastic effect which produces variability in the stimulus effects based on prior stimulation, it is a shame that the variability in neural firing and force induced skin displacements have been presented, and are similarly variable, but there has been no investigation of a link between the two. I believe with these data the authors can go beyond demonstrating shared variability. The force per se is clearly not faithfully represented in the neural signal, being masked by stimulation history, and it is of interest if the underlying resultant contact mechanics are.

Validity of conclusions:

The authors have succeeded in demonstrating skin viscoelasticity has an impact on skin contact mechanics with a given force and that this impacts on the resultant neural coding of force. Their study has been well designed and the results support their conclusions. The importance and scope of the work is adequately outlined for readers to interpret the results and significance.

Impact:

This study will have important implications for future studies performing tactile stimulation and evaluating tactile feedback during motor control tasks. In detailed studies of tactile function, it illustrates the necessity to measure skin contact dynamics to properly understand the effects of a force stimulus on the skin and mechanoreceptors.

---

## [Author Response]

The following is the authors’ response to the original reviews

**Public Reviews:**

**Reviewer #1 (Public Review):**
The authors investigate how the viscoelasticity of the fingertip skin can affect the firing of mechanoreceptive afferents and they find a clear effect of recent physical skin state (memory), which is different between afferents. The manuscript is extremely well-written and well-presented. It uses a large dataset of low threshold mechanoreceptive afferents in the fingertip, where it is particularly noteworthy that the SA-2s have been thoroughly analyzed and play an important role here. They point out in the introduction the importance of the non-linear dynamics of the event when an external stimulus contacts the skin, to the point at which this information is picked up by receptors. Although clearly correlated, these are different processes, and it has been very well-explained throughout. I have some comments and ideas that the authors could think about that could further improve their already very interesting paper. Overall, the authors have more than achieved their aims, where their results very much support the conclusions and provoke many further questions. This impact of the previous dynamics of the skin affecting the current state can be explored further in so many ways and may help us to better understand skin aging and the effects of anatomical changes of the skin.At the beginning of the Results, it states that FA-2s were not considered as stimuli did not contain mechanical events with frequency components high enough to reliably excite them. Was this really the case, did the authors test any of the FA-2s from the larger dataset? If FA-2s were not at all activated, this is also relevant information for the brain to signal that it is not a relevant Pacinian stimulus (as they respond to everything). Further, afferent receptive fields that were more distant to the stimulus were included, which likely fired very little, like the FA-2s, so why not consider them even if their contribution was low?

Thank you for bringing this up, we have now clarified in the text that while FA-2s did respond at a low rate during the experiment, their responses were not reliably driven by the force stimuli. In the Methods section we have included the following text:

“Initially, 10 FA-2 neurons were also included in the analysis. But their responsiveness during the experiment was remarkably low, and unlike the other neuron types, their responses were rarely affected by force stimuli. Specifically, only one of the observed FA-2 neurons responded during the force protraction phases. Due to the lack of clear stimulus-driven responses, FA-2 neurons were subsequently excluded from further analysis.”

One question that I wondered throughout was whether you have looked at further past history in stimulation, i.e. not just the preceding stimulus, but 2 or 3 stimuli back? It would be interesting to know if there is any ongoing change that can be related back further. I do not think you would see anything as such here, but it would be interesting to test and/or explore in future work (e.g. especially with sticky, forceful, or sharp indentation touch). However, even here, it could be that certain directions gave more effects.

This is a very interesting question! A discernible effect from the previous stimulus could persist at the end of the current stimulation (see Figure 4C), potentially influencing the next one—a 2-stimuli-back effect. Unfortunately, our experimental design did not allow for rigorous testing of this effect. While all possible pairs of stimulus directions were included in immediately consecutive trials, this was not the case for pairs separated by additional trials. Hence, the combination of a likely weak effect and limited variation in history precluded a thorough analysis of a 2-stimuli-back effect. Future work should delve into the time course of the viscoelastic effect in greater detail.

Did the authors analyze or take into account the difference between receptive field locations? For example, did afferents more on the sides have lower responses and a lesser effect of history?

An investigation into the potential impact of the relationship between the receptive field location on the fingertip skin and the primary contact site of the stimulus surface revealed no discernible influence for SA-1 and SA-2 neurons. In contrast, FA-1 neurons, particularly those predominantly sensitive to the previous stimulation or displaying mixed sensitivity, exhibited a tendency to terminate near the primary stimulation site. We have added these observations to the text:

“We found no straightforward relationship between a neuron's sensitivity to current and previous stimulation and its termination site in fingertip skin. Specifically, there was no statistically significant effect of the distance between a neuron's receptive field center and the primary contact site of the stimulus surface on whether neurons signaled current, prior, or mixed information for SA-1 (Kruskal-Wallis test H(2)=3.86, p = 0.15) or SA-2 neurons (H(2)=0.75, p=0.69). However, a significant difference emerged for FA-1 neurons (H(2)=8.66, p=0.01), indicating that neurons terminating closer to the stimulation site on the flat part of the fingertip were more likely to signal past or mixed information.”

Was there anything different in the firing patterns between the spontaneous and non-spontaneously active SA-2s? For example, did the non-spontaneous show more dynamic responses?

The firing patterns of both spontaneously and non-spontaneously active SA-2 neurons shared similarities in terms of adaptation and range of firing rate modulation in response to force stimuli, i.e., ‘dynamic response’. The distinction lay in the pattern of modulation of the firing rate associated with stimulus presentations. For spontaneously active SA-2 neurons, this modulation occurred around a significant background discharge, implying that a force stimulus could either decrease or increase the firing rate, depending on how it deformed the fingertip. This characteristic is well illustrated by the firing pattern of the neuron depicted in the lower panels of Figure 3D. Conversely, in non-spontaneously active SA-2 neurons, a force stimulus could only induce an increase in the firing rate or no change. Although the neuron depicted in the upper panels of Figure 3D exhibited some background activity, it serves to exemplify this characteristic. In the text, we have elucidated the dynamics of the SA-2 neuron response by highlighting that force stimulation can either decrease or increase the firing rate in neurons with spontaneous activity through the following addition/change:

“This increased variability was most evident during the force protraction phase where most neurons exhibited the most intense responses. Increased variability was also observed in instances where the dynamic response to force stimulation involved a decrease in the firing rate (lower panels of Figure 3D). This phenomenon was observed in SA-2 neurons that maintained an ongoing discharge during intertrial periods (cf. Fig. 2A). In these cases, the response to a force stimulus constituted a modulation of the firing rate around the background discharge, signifying that a force stimulus could either decrease or increase the firing rate depending on the prevailing stimulus direction.”

Were the spontaneously active SA-2 afferents firing all the time or did they have periods of rest - and did this relate to recent stimulation? Were the spontaneously active SA-2s located in a certain part of the finger (e.g. nail) or were they randomly spread throughout the fingertip? Any distribution differences could indicate a more complicated role in skin sensing.

SA-2 neurons, in general, are well-known for undergoing significant post-stimulation depression (e.g., Knibestöl and Vallbo, 1970; Chambers et al., 1972; Burgess and Perl, 1973). In our force stimulations, this post-excitatory depression manifested as a reduced or absent response during the latter part of the stimulus retraction period for stimuli in directions that markedly excited the neuron. The excitability recovered when the fingertip relaxed during the subsequent intertrial period, and for "spontaneously active" neurons, the firing resumed (see examples in Figure 7A). Furthermore, some “spontaneously active” neurons could be silenced or exhibit a near-silent period during force stimulation for certain force directions, while the spontaneous firing returned during the upcoming intertrial period when the fingertip shape recovered (for example, see responses to stimulation in the proximal and especially ulnar directions in the top panel in Figure 7A).

Regarding the location of the receptive field centres of spontaneously active and non-spontaneously active SA-2 neurons on the fingertip we did not observe any obvious spatial segregation. To illustrate this, we have revised Figure 1A by color-marking SA-2 neurons that exhibited ongoing activity in intertrial periods, and the figure caption has been modified accordingly:

“Figure 1. Experimental setup. A. Receptive field center locations shown on a standardized fingertip for all first-order tactile neurons included in the study, categorized by neuron type. Purple symbols denote spontaneously active SA-2 neurons exhibiting ongoing activity without external stimulation.”

Did the authors look to see if the spontaneous firing in SA-2s between trials could predict the extent to which the type 1 afferents encode the proceeding stimulus? Basically, does the SA-2 state relate to how the type 1 units fire?

We found no clear indications that the responses of FA-1 and SA-1 could be readily anticipated based on the firing patterns of SA-2 neurons.

In the discussion, it is stated that "the viscoelastic memory of the preceding loading would have modulated the pattern of strain changes in the fingertip differently depending on where their receptor organs are situated in the fingertip". Can the authors expand on this or make any predictions about the size of the memory effect and the distance from the point of stimulation?

We have explored this topic further in the text, referring to recent studies modeling essential aspects of fingertip mechanics. However, in our view, current models lack the capability to predict the specific nature sought by the reviewer. These models should include a detailed understanding of the intricate networks of collagen fibers anchoring the pulp tissue at the distal phalangeal bone and the nail. They should also consider potential inherent directional preferences of the receptor organs, attributed to their microanatomy. The text modifications are as follows:

“In addition to the receptor organ locations, the variation in sensitivity among neurons to fingertip deformations in response to both previous and current loadings would stem from the fingertip’s geometry and its complex composite material properties. Possible inherent directional preferences of the receptor organs, attributed to their microanatomy, could also be significant. However, mechanical anisotropy, particularly within the viscoelastic subcutaneous tissue of the fingertip induced by intricately oriented collagen fiber strands forming fat columns in the pulp (Hauck et al., 2004), are likely to play a crucial role. This anisotropy would shape the dynamic pattern of strain changes at neurons' receptor sites, intricately influencing a neuron's sensitivity not only to current but also to preceding loadings. Indeed, recent modeling efforts suggest that such mechanical anisotropy strongly influences the spatiotemporal distribution of stresses and strains across the fingertip (Duprez et al., 2024).”

Relatedly, we have included additional text to provide a more comprehensive explanation of the “bulk deformation” of the fingertip that occurs during the loadings:

“As pressure increases in the pulp, the pulp tissue bulges at the end and sides of the fingertip. Simultaneously, the tangential force component amplifies the bulging in the direction of the force while stretching the skin on the opposite side.”

In the discussion, it would be good if the authors could briefly comment more on the diversity of the mechanoreceptive afferent firing and why this may be useful to the system.

The diversity in responses among neurons is instrumental in enhancing the information transmitted to the brain by averting redundancy in information acquisition. This diversity thereby contributes to an overall increase in information. We've included a brief statement, along with several references, underscoring this concept:

"The resulting diversity in the sensitivities of neurons might enhance the overall information collected and relayed to the brain by the neuronal population, facilitating the discrimination between tactile stimuli or mechanical states of the fingertip (see Rongala et al., 2024; Corniani et al., 2022; Tummala et al., 2023, for more extensive explorations of this idea)."

Also, the authors could briefly discuss why this memory (or recency) effect occurs - is it useful, does it serve a purpose, or it is just a by-product of our skin structure? There are examples of memory in the other senses where comparisons could be drawn. Is it like stimulus adaptation effects in the other senses (e.g. aftereffects of visual motion)?

We have expanded the concluding paragraph of the discussion, specifically delving into the question of whether the mechanical memory effect serves a deliberate purpose or is simply an incidental byproduct of our skin structure:

“In any case, the viscoelastic deformability of the fingertips plays a pivotal role in supporting the diverse functions of the fingers. For example, it allows for cushioned contact with objects featuring hard surfaces and allows the skin to conform to object shapes, enabling the extraction of tactile information about objects' 3D shapes and fine surface properties. Moreover, deformability is essential for the effective grasping and manipulation of objects. This is achieved, among other benefits, by expanding the contact surface, thereby reducing local pressure on the skin under stronger forces and enabling tactile signaling of friction conditions within the contact surface for control of grasp stability. Throughout, continuous acquisition of information about various aspects of the current state of the fingertip and its skin by tactile neurons is essential for the functional interaction between the brain and the fingers. In light of this, the viscoelastic memory effect on tactile signaling of fingertip forces can be perceived as a by-product of an overall optimization process within prevailing biological constraints.”

One point that would be nice to add to the discussion is the implications of the work for skin sensing. What would you predict for the time constant of relaxation of fingertip skin, how long could these skin memory effects last? Two main points to address here may be how the hydration of the skin and anatomical skin changes related to aging affect the results. If the skin is less viscoelastic, what would be the implications for the firing of mechanoreceptors?

It is likely that the time constant depends to some extent on mechanical factors of the skin, which will likely change due to age or environmental factors. However, while these questions are intriguing, they fall outside the scope of the current study and we are not aware of studies that have addressed these issues directly in experiments either.

How long does it take for the effect to end? Again, this will likely depend on the skin's viscoelasticity. However, could the authors use it in a psychophysical paradigm to predict whether participants would be more or less sensitive to future stimuli? In this way, it would be possible to test whether the direction modifies touch perception.

Time constants for tissue viscoelasticity have been estimated to extend up to several seconds (see citations in the introduction). While direct perceptual effects could indeed be explored through psychophysical experimental paradigms, we are currently unaware of any studies specifically addressing the type of effect described in this study. In addition to the statement that, concerning manipulation and haptic tasks, "to our knowledge, a possible influence of fingertip viscoelasticity on task performance has not been systematically investigated," we have now also addressed tactile psychophysical tasks conducted during passive touch with the following sentence in the text:

“Similarly, there is a lack of systematic investigation of potential effects of fingertip viscoelasticity on performance in tactile psychophysical tasks conducted during passive touch.”

**Reviewer #2 (Public Review):**
Summary:The authors sought to identify the impact skin viscoelasticity has on neural signalling of contact forces that are representative of those experienced during normal tactile behaviour. The evidence presented in the analyses indicates there is a clear effect of viscoelasticity on the imposed skin movements from a force-controlled stimulus. Both skin mechanics and evoked afferent firing were affected based on prior stimulation, which has not previously been thoroughly explored. This study outlines that viscoelastic effects have an important impact on encoding in the tactile system, which should be considered in the design and interpretation of future studies. Viscoelasticity was shown to affect the mechanical skin deflections and stresses/strains imposed by previous and current interaction force, and also the resultant neuronal signalling. The result of this was an impaired coding of contact forces based on previous stimulation. The authors may be able to strengthen their findings, by using the existing data to further explore the link between skin mechanics and neural signalling, giving a clearer picture than demonstrating shared variability. This is not a critical addition, but I believe would strengthen the work and make it more generally applicable.Strengths:- Elegant design of the study. Direct measurements have been made from the tactile sensory neurons to give detailed information on touch encoding. Experiments have been well designed and the forces/displacements have been thoroughly controlled and measured to give accurate measurements of global skin mechanics during a set of controlled mechanical stimuli.- Analytical techniques used. Analysis of fundamental information coding and information representation in the sensory afferents reveals dynamic coding properties to develop putative models of the neural representation of force. This advanced analysis method has been applied to a large dataset to study neural encoding of force, the temporal dynamics of this, and the variability in this.Weaknesses:- Lack of exploration of the variation in neural responses. Although there is a viscoelastic effect that produces variability in the stimulus effects based on prior stimulation, it is a shame that the variability in neural firing and force-induced skin displacements have been presented, and are similarly variable, but there has been no investigation of a link between the two. I believe with these data the authors can go beyond demonstrating shared variability. The force per se is clearly not faithfully represented in the neural signal, being masked by stimulation history, and it is of interest if the underlying resultant contact mechanics are.

Thank you for this suggestion. We have added a new section investigating the link between skin deformation and neural firing in more depth via a simple neural model. Please see our answer below in the ‘Recommendations’ section for further details.

Validity of conclusions:The authors have succeeded in demonstrating skin viscoelasticity has an impact on skin contact mechanics with a given force and that this impacts the resultant neural coding of force. Their study has been well-designed and the results support their conclusions. The importance and scope of the work is adequately outlined for readers to interpret the results and significance.Impact:This study will have important implications for future studies performing tactile stimulation and evaluating tactile feedback during motor control tasks. In detailed studies of tactile function, it illustrates the necessity to measure skin contact dynamics to properly understand the effects of a force stimulus on the skin and mechanoreceptors.
**Recommendations for the authors:**

**Reviewer #1 (Recommendations For The Authors):**
(Very) minor comments- The authors say at the beginning of the Results that, "The fourth type of tactile neurons in the human glabrous skin, fast adapting type II neurons...". Although generally written that there are four types of afferent in the glabrous skin, it would be better to state that these are low-threshold A-beta myelinated mechanoreceptive afferents, at least one time, as there are other types of afferent in the glabrous skin that respond to mechanical stimulation (e.g. low and high threshold C-fibers).

This is now clarified at the start of the Results section:

“We recorded action potentials in the median nerve of individual low-threshold A-beta myelinated first-order human tactile neurons innervating the glabrous skin of the fingertip…”

- Fig. 3: Could you add '(N)' as the measurement of force for Fig. 3A for Fz, Fy, and Fz? Also, please change 'Data was recorded' to 'Data were recorded' in the legend.

Fixed.

- At the beginning of the Methods, you say that your study conforms to the Declaration of Helsinki, which actually requires pre-registration in a database. If you did not pre-register your study, please can you add '... in accordance with the Declaration of Helsinki, apart from pre-registration in a database'.

Thanks for making us aware of this. We have added the suggested qualifier to the ethics statement.

**Reviewer #2 (Recommendations For The Authors):**
The neural representation/encoding of the actual displacement vectors would be a useful addition to the analyses. These vectors have been demonstrated to systematically change with the condition in the irregular series (Figure 2E) and will thus significantly act on the dynamics of induced mechanical changes in the skin with a given interaction force. Thus, it could be examined how the neurons code the magnitude of displacements as well as their direction. An evaluation of the extent to which the imposed displacement magnitudes are encoded in the neural responses would be a useful addition in explaining the signalling of the force events and how the central nervous system decodes these. Evaluating an alternative displacement encoding for comparison to pure force encoding may reveal more about how contact events are represented in the tactile system, which must decode these variable afferent signals to reconstruct a percept of the interaction. It could then be explored how the central nervous system may then scale the dynamic afferent responses based on the background viscoelastic state likely to be present in the SA-II afferent signals (Figure 7) for a context in which to evaluate the dynamic contact forces. This may of course be a complex relationship for the type-I afferents, where the underlying mechanical events evoking the firing (microslips not represented in global forces) have not been measured here. Such a model could be more widely applicable, as the skin viscoelasticity and displacement magnitudes are a straightforward measurement metric and could perhaps be used as a better proxy for neural signalling. This would allow the investigation of a wider variety of forces, and the study of the timing of the viscoelastic effect, both of which have been fixed here. This would give the work a broader impact, rather than just highlighting that this effect produces variability, it could reveal if this mechanical feature is structured in the neural representation. The categorical encoding/decoding tested here is specific to the stimuli used (magnitudes, intervals), but there is the possibility that this may be more generally applicable (within the bounds of forces/speeds) if the underlying basis of the variability in the signalling produced by the viscoelasticity is identified. Since the time course of the viscoelasticity has not been measured here (fixed forces and intervals), further study is required to fully understand the implications this has for a wider variety of situations.

We agree that a better understanding of how the mechanical deformations are reflected in the resulting spike trains would be valuable. While ultimately a full understanding will need precise measurements of skin deformation across the whole fingertip to account for mechanical propagation to mechanoreceptor locations, relating the deformations at the contact location with neural firing patterns directly can provide useful hints into which aspects of deformation are encoded and how. To this end, we ran a new analysis that aimed to predict the time-varying neural responses directly from the recorded mechanical movements of the contactor.

Below we have reproduced the new results and methods text along with the additional figures for this analysis. Note that we have also added text in the Discussion to interpret these findings in the context of our other results.

New section in Results titled Predicting neural responses from contactor movements: “The similarity in the history-dependent variation in neural firing and fingertip deformation at a given force stimulus suggests that neuronal firing is determined by how the fingertip deforms rather than the applied force itself. However, this similarity does not clarify the relationship between fingertip deformation dynamics and neural signaling. To investigate further, we fit cross-validated multiple linear regression models to evaluate how well distinct aspects of contactor movement could predict the time-varying firing rates of individual neurons during the protraction phases of the irregular sequence. The models used predictors based on (1) the three-dimensional position of the contactor, (2) its three-dimensional velocity, (3) a combination of position and velocity signals, and, finally, (4) position and velocity signals along with all possible two-way interactions between them, capturing potentially complex relationship between fingertip deformations and neural signaling.

Comparing the variance explained (R^2^) by each regression model for each neuron type revealed clear differences between the models (Figure 5A). A two-way mixed design ANOVA, with regression model as within-group effects and neuron type as a between-group effect revealed a main effect of model on variance explained (F(3,462) = 815.5, p < 0.001, η_p_^2^ = 0.84). Model prediction accuracy overall increased with the number of predictors, with the two-way interaction model outperforming all others (p < 0.001 for all comparisons, Tukey’s HSD). Additionally, a significant main effect of neuron type (F(2,154) = 29.8, p < 0.001, η_p_^2^ = 0.28) and a significant interaction between regression model and neuron type were observed (F(6,462) = 50.8, p < 0.001, η_p_^2^ = 0.40).

For neuron type, model predictions were most accurate for SA-2 neurons, followed by SA-1 neurons, with FA-1 neurons showing the lowest accuracy (p < 0.003 for all comparisons, Tukey’s HSD). The interaction between model and neuron type revealed distinct patterns. For SA-1 and SA-2 neurons, position-only and velocity-only models had similar prediction accuracy (p ≥ 0.996, Tukey’s HSD) with no significant differences between these neuron types (p ≥ 0.552, Tukey’s HSD). FA-1 neurons performed poorly with the position-only model but showed higher accuracy with the velocity-only model (p < 0.001, Tukey’s HSD) and better than SA-1 neurons (p = 0.006, Tukey’s HSD). Models combining position and velocity predictors (without interactions) surpassed both position-only and velocity-only models for SA-1 and SA-2 neurons (p < 0.001, Tukey’s HSD). Overall, the differences between neuron types broadly match their tuning to static and dynamic stimulus properties.

The two-way interaction model, accounting for most variance in neural responses, produced mean R^2^ values of 0.75 for FA-1, 0.88 for SA-1, and 0.91 for SA-2 neurons (Figure 5A). To evaluate the contribution of the different predictors, we ranked them using the permutation feature importance method, focusing on the six most important ones. Regression analyses using only these variables explained almost all of the variance explained by the full model, with a median R^2^ reduction of just 0.055 across all neurons. Across all neuron types, at least half included all three velocity components (dPx, dPy, dPz) among the top six, with FA-1 neurons showing the highest prevalence (Figure 5B). Interactions between normal position (Pz) and each velocity component were also frequently observed, while interactions involving tangential position and velocity components were less common. Interactions among velocity components were relatively well represented, followed by interactions limited to position components. Position signals were generally less represented, except for normal position (Pz) in slowly adapting neurons, where it appeared in 50% of SA-1 and 68% of SA-2 neurons. Despite these broad trends, important predictors varied widely across ranks even within a given neuron class (see Figure 5-figure supplement 1), and even the most frequent variables appeared in only a subset of cases, suggesting broad variability in sensitivity across neurons.”

New methods paragraph titled Predicting time-varying firing rates from skin deformations:

“This analysis was conducted in Python (v3.13) with pandas for data handling, numpy for numerical operations, and scikit-learn for model fitting and evaluation.

To assess how well individual neurons' time-varying firing rates could be predicted from simultaneous contactor movements, we fitted multiple linear regression models (see Khamis et al., 2015, for a similar approach). This analysis focused on the force protraction phase of the irregular sequence, where neurons were most responsive and sensitive to stimulation history. Data from 100 ms before to 100 ms after the protraction phase (between -0.100 s and 0.225 s relative to protraction onset) were included for each trial. Neurons were included if they fired at least two action potentials during the force protraction phase and the following 100 ms in at least five of the 25 trials. This ensured sufficient variability in firing rates for meaningful regression analysis, resulting in 68 SA-1, 38 SA-2, and 51 FA-1 neurons being included.

Contractor position signals digitized at 400 Hz were linearly interpolated to 1000 Hz. Instantaneous firing rates, derived from action potentials sampled at 12.8 kHz, were resampled at 1000 Hz to align with position signals. A Gaussian filter (σ = 10 ms, cutoff ~16 Hz) was applied to the firing rate as well as to the position signals before differentiation. To account for axonal conduction (8–15 ms) and sensory transduction delays (1–5 ms), firing rates were advanced by 15 ms to align approximately with independent variables.

Regressions were performed using scikit-learn's Ridge and RidgeCV regressors, which apply L2 regularization to mitigate overfitting. Hyperparameter tuning for the regularization parameter (alpha) was performed using GridSearchCV with a predefined range (0.001–1000.0), incorporating five-fold cross-validation to select the best value. To minimize overfitting risks, model performance was further validated with independent five-fold cross-validation (KFold), and R^2^ scores were computed using cross_val_score.

We constructed four linear regression models with increasing complexity: (1) Position-only, using three-dimensional contactor positions (Px, Py, Pz); (2) Velocity-only, using three-dimensional velocities (dPx, dPy, dPz); (3) Combined, including all position and velocity signals (6 predictors); and (4) Interaction, including all signals and their two-way interactions (21 predictors). All features were standardized using StandardScaler to improve regularization and model convergence. PolynomialFeatures generated second-order interaction terms for the interaction model. Feature importance was evaluated with permutation_importance, and simpler models were built using the most important features. These models were validated through cross-validation to assess retained explanatory power.”

Minor:- It would be useful to add a brief description of the material aspects of the contactor tip to the methods (as per Birznieks 2001).

We have added the following statement:

“To ensure that friction between the contactor and the skin was sufficiently high to prevent slips, the surface was coated with silicon carbide grains (50–100 μm), approximating the finish of smooth sandpaper.”

- The axes labelling on Figure 3A and legend description is ambiguous, probably placing the Px, Py, and Pz labels on the far left axes and the Fx, Fy, and Fz on the right side of the far right axes would make this clearer.

Label placement has been improved along with some other minor fixes.

- For the quasi-static phase analysis, the phrase "absence of loading" used in reference to the interstimulus period and SA-II afferents does not seem to be a correct description. The finger is still loaded (at least in the normal direction), with a magnitude of imposed displacement that counteracts the viscoelastic force exerted by the skin mechanics of the fingertip. Although there is a zero net-force load, a mechanical stimulus is still being actively applied to the skin.

We have changed the wording throughout the text and now consistently refer either to the “interstimulus period” directly or to an “absence of externally applied stimulation” to avoid confusion.